# Development and content of a community-based reablement programme (I-MANAGE): a co-creation study

Ines Mouchaers [1,2,3] Hilde Verbeek,[1,2] Gertrudis I J M Kempen,[1,2] Jolanda C M van Haastregt,[1,2] Ellen Vlaeyen,[4,5] Geert Goderis,[3] Silke F Metzelthin[1,2]

For numbered affiliations see end of article.

**Correspondence to**
Ines Mouchaers;
i.mouchaers@
maastrichtuniversity.nl

## ABSTRACT

**Objectives** As age increases, people generally start experiencing problems related to independent living, resulting in an increased need for long-term care services. Investing in sustainable solutions to promote independent living is therefore essential. Subsequently, reablement is a concept attracting growing interest. Reablement is a person-centred, holistic approach promoting older adults' active participation through daily, social, leisure and physical activities. The aim of this paper is to describe the development and content of I-MANAGE, a model for a reablement programme for community-dwelling older adults.

**Design** The development of the programme was performed according to the Medical Research Council framework as part of the TRANS-SENIOR international training and research network. A co-creation design was used, including literature research, observations, interviews, and working group sessions with stakeholders.

**Setting and participants** The interviews and working group sessions took place in the Dutch long-term home care context. Stakeholders invited to the individual interviews and working group sessions included care professionals, policymakers, client representatives, informal caregiver representatives, informal caregivers, and scientific experts.

**Results** The co-creation process resulted in a 5-phase interdisciplinary primary care programme, called I-MANAGE. The programme focuses on improving the self-management and well-being of older adults by working towards their meaningful goals. During the programme, the person's physical and social environment will be put to optimal use, and sufficient support will be provided to informal caregivers to reduce their burden. Lastly, the programme aims for continuity of care and better communication and coordination.

**Conclusion** The I-MANAGE programme can be tailored to the local practices and resources and is therefore suitable for the use in different settings, nationally and internationally. If the programme is implemented as described, it is important to closely monitor the process and results.

## STRENGTHS AND LIMITATIONS OF THIS STUDY

⇒ Intervention development was conducted using a systematic approach based on the Medical Research Council framework.
⇒ The intervention and its development is theory and evidence based as a result of extensive literature research.
⇒ All end users were represented during the development process, increasing its acceptability and feasibility.
⇒ Data triangulation was used during the development process, increasing validity of the results.
⇒ Client and informal caregiver representatives were included in the working group instead of clients and informal caregivers themselves, possibly missing an important voice.

## BACKGROUND

The amount of older adults experiencing disabilities will increase over time and, while a large proportion of the older population remains independent, others will experience an increased need for support.[1] Moreover, 50% of people aged 85 years or older require care and/or support with daily activities.[2] As a result, it is expected that their demands for long-term care services will increase. When older adults live in an environment that is unsafe and does not meet their needs, the challenges they might face regarding independent living will increase further.[3] Previous research showed that a maladjusted environment negatively affects disability, which could lead to an accumulation of health risks, loss of independence, poor quality of life (QoL) and depression.[4–6] Care and service delivery in the community is often fragmented, with little coordination and poor communication among care providers, clients and informal caregivers.[3 7–9] Furthermore, the focus of care is often on eliminating specific diseases and symptoms instead of supporting the

remaining capacity to maintain QoL and independent living.[10 11] Rather than performing tasks *with* their clients, care professionals often tend to take over.[11] Failing to properly tackle these challenges could increase the use of health and social care and related costs.[12] In addition, this could lead to unnecessary (re)hospitalisations or permanent nursing home placement, which each have their own risks (eg, increased mortality)[13] and at a time when financial and workforce resources are shrinking.[7 12 14] It is therefore essential to invest in sustainable solutions to promote independent living.[15–17]

A concept attracting growing interest in promoting independent living among older adults is reablement. This is a person-centred, holistic approach promoting active participation of older adults in daily activities through social, leisure and physical activities chosen by the older adult in line with their preferences, either at home or in the community.[18] Instead of creating dependency by taking over tasks, care professionals identify the capabilities and opportunities of individuals to maximise their independence by supporting them to achieve their goals, through participation in daily activities, home modifications, assistive devices and involvement of their social network.[11 18–20] Current evidence on the effectiveness of reablement interventions is inconclusive,[11] however, several systematic reviews have indicated the positive results of reablement relating to activities of daily living (ADL) functioning and health-related QoL.[21–23] Due to the promising results, interest in implementing reablement into everyday care is growing internationally. In Denmark, New Zealand and the UK, reablement has more-or-less been successfully implemented across the whole country.[1] For example, in Denmark reablement in long-term care for older adults was legally introduced in 2015, meaning that all municipalities must offer reablement interventions and all applicants for home care are assessed for potential for reablement before being offered conventional home care.[2]

Despite the promising results and successful implementation abroad, contextual differences mean this is no indication that it would necessarily be effective in its current format in the Dutch home care setting.[24] Implementing reablement is a complex process and influenced by multiple factors, such as organisational factors, individual and social attitudes towards a new form of care, technological factors related to communication and financial factors.[25 26] It is crucial to critically consider these factors in the design, delivery and evaluation of reablement.[27] Moreover, to implement reablement in the Dutch home care setting, existing programmes need to be revised and adapted to suit the current context, which is crucial when developing and delivering complex healthcare interventions.[27] However, the development and content of community care programmes, such as reablement programmes, are often insufficiently described in the scientific literature.[19 28–30] There are only limited articles available that describe either the development of the programme or its content in detail[31–33] and these descriptions are often included as part of a feasibility or pilot study.[29] This offers little guidance to replicate or build on the previous findings of such programmes, despite this being essential for the development of new and implementation of existing programmes in different settings.[29 30 34]

This paper therefore describes the development, using a co-creation process, and content of I-MANAGE, a reablement programme for community-dwelling older adults to improve older adults' self-management and participation in daily life, while also increasing QoL and decreasing informal caregivers' burden. By describing the development and content of the programme in detail, we increase the replicability and prevent other researchers from reinventing the wheel. The programme is specifically suitable for the Dutch community care setting, however, due to its extensive description, this manuscript may also provide a model for implementation in other countries.

## METHODS

To describe the development process of the programme and ensure completeness of reporting, we used the guidance for reporting for intervention development studies in health research (online supplemental file 1).[29] This checklist provides a clear and structured basis for the reporting of programme development, as well as the description of the content of the programme.

### Design

I-MANAGE was developed between September 2019 and June 2021 using a co-creation design. The programme was developed following the first phase (development) of the Medical Research Council (MRC) framework for the development and evaluation of complex interventions.[35 36]

### Patient and public involvement

Co-creation was initiated by the researchers as a response to the challenges that are caused by an ageing society (eg, increasing care needs, decreasing staff). To deal with these challenges the Dutch government stimulates an ageing in place policy and promotes another way of (home) care delivery moving from 'doing for…' towards 'doing with…' clients, or in the best case to enable clients to do things by themselves again. These developments are also adopted by Dutch care organisations, which hope to improve the quality and sustainability of their care services.[37] All end users (care professionals who would be implementing and delivering the programme, as well as the target population) were represented and involved during the development process of the programme as members of the working groups and by participating in the individual interviews. However, they were not involved in the development of the study design of dissemination of the findings.

## Setting and participants

I-MANAGE is based on international evidence and tailored to the Dutch home care context. Home care in the Netherlands includes personal care (ie, assistance with ADL), nursing care (ie, medical assistance) and domestic support (ie, assistance with instrumental ADL (IADL)).[38] Usually home care is funded by two statutory forms of insurance cover care: the *Health Insurance Act* (ZVW) and the *Social Support Act* (WMO). Clients often use a combination of ZVW (eg, general practitioner care, therapists, hospital care or medication), and WMO (eg, domestic support, home adaptations).[38 39] The programme was developed for community-dwelling older adults, irrespective of age or cognitive and functional status.

The programme was developed in co-creation with stakeholders who participated in observations, individual interviews and working groups. Online supplemental file 2 provides an overview of all stakeholders involved and the research activities they participated in. Participants were recruited from the professional network of the researchers. They were informed about the study and asked to participate via email. When participants agreed to participate, verbal or written informed consent was provided before the start of each interview or working group session.

## Data collection

First, a logic model was created. A logic model is a tool to illustrate how a programme will create change.[40] The logic model systematically visualises the aim and subaims of the programme, the programme components and the intended outcomes.[40] The logic model was developed using six iterative programme development steps: (1) identifying the problem, (2) identifying the evidence, (3) identifying or developing a theory, (4) determining needs, (5) examining current practices and the context and (6) modelling processes and outcomes.[41] The logic model was then translated to practice (step 7). Data collection was performed following a non-linear and iterative process as described by Bleijenberg *et al*,[41] and by using a variety of data collection methods, including literature research, observations, individual interviews and working groups. Figure 1 illustrates the development process in detail.

### Development of a logic model (steps 1–6)

All steps were guided by literature research, especially steps 2 and 3 of the development process. To gain insight into current evidence-based practices (step 2), the relevant scientific literature on reablement programmes was reviewed. The methodology and results of this literature research is described elsewhere.[42] To identify a theoretical foundation for the programme (step 3), a literature review was performed on the concept of disability and its underlying causal mechanisms.[43]

Furthermore, a working group was composed and invited to three sessions (1.5 hours each). Members of the working group included physiotherapists (PT),

occupation therapists (OT), registered nurses (RN), a domestic support worker (DSW), a policymaker of the local municipality, a psychologist, a client representative, informal caregiver representatives and a geriatric rehabilitation expert. The first session focused on the identification of the problem (step 1) and the examination of the current practice and context (step 5). The second session focused on the previously identified evidence, which participants could complement with practices from their own field of work (step 2), and the determination of needs (step 4), closely related to the previously identified problems. The third session focused on modelling the process and outcomes (step 6), during which a preliminary logic model was presented, on which participants could provide feedback. Between sessions, working group members were consulted for additional input and clarification if needed. The researchers processed the results from each session and the additional information in order to be used as a starting point for the next session. The working group sessions were led by the first and last author (IM and SM).

We then interviewed two PTs, a policymaker of the local municipality, a client and informal caregiver representative and three experts in the field of geriatric rehabilitation research. In total, eight interviews were conducted, all focusing on identifying the problem (step 1), determining the needs (step 4) and gaining insight into current practice and context (step 5). In addition, we interviewed two informal caregivers. Topics addressed during these interviews included their role as informal caregiver, how involved they were in the care process, how they were or felt supported and by whom and what they would like to change in the process.

Lastly, observations were performed by the researchers. Observations were conducted in the traditional community care setting to examine current practice and context of home care services (step 5) and to identify problems (step 1). Six observations, each lasting half a day, were conducted by shadowing a DSW, an RN and a nursing assistant, and 3 days were spent with allied health professionals. Field notes were taken throughout the observation periods.

### Translation to practice (step 7)

For each of the six programme components in the logic model, we invited members of the working group, a geriatrician, informal care consultant, reablement researcher, informal caregiver and a community care teams' manager to a session to translate components into practice (step 7). Based on the programme component, the most relevant stakeholders were invited. For example, on informal caregiver support, we invited an informal care consultant, an informal caregiver and a psychologist; resulting in three to four members per working group. Each working group attended one session lasting 1 hour. Each session began with an introduction to the goal of the programme, the logic model and the results of steps 1–6 related to the programme component. Afterwards,

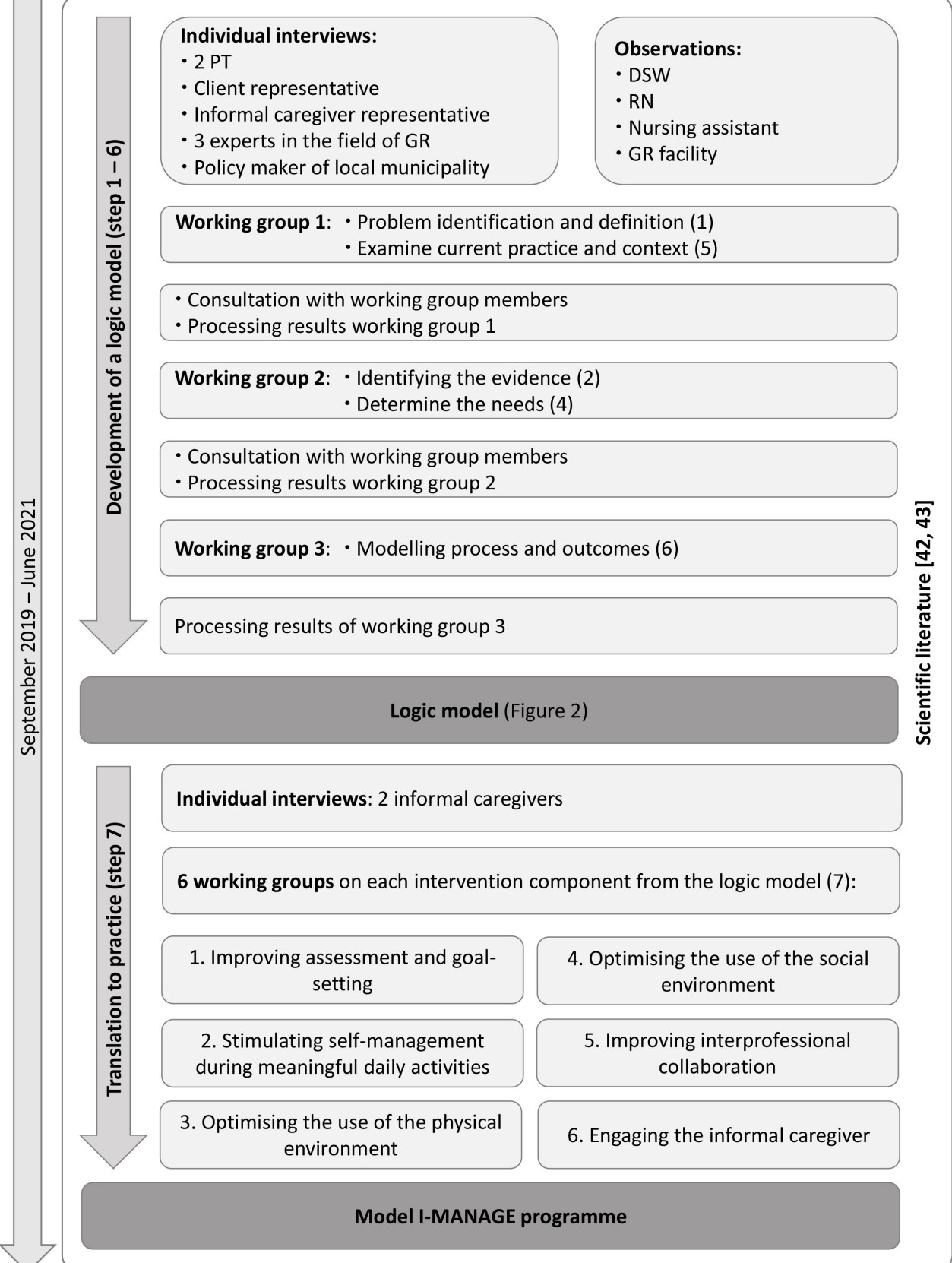

**Figure 1** The development process over time. This figure presents the development process over time from September 2019 until June 2021. The whole process is guided by scientific literature. The figure presents the two parts of the development process each with their related activities. The dark grey rectangles present the (intermediate) result of each part of the process. The number in between brackets represent the seven steps of the development phase as described by Bleijenberg et al.[41] DSW, domestic support worker; GR, geriatric rehabilitation; PT, physiotherapist; RN, registered nurse.

**Table 1** Overview of the different methodologies used in each development step defined by Bleijenberg et al[41]

| Programme development steps | Methodology | | | |
| --- | --- | --- | --- | --- |
| | Literature research | Observations | Individual interviews | Working groups |
| 1. Problem identification | ◍ | ● | ● | ● |
| 2. Identifying the evidence | ● | | | ● |
| 3. Identifying or developing theory | ● | | | |
| 4. Determine the needs | ◍ | | ● | ● |
| 5. Examining current practice and context | ◍ | ● | ● | ● |
| 6. Modelling process and outcomes | ◍ | ◍ | ◍ | ● |
| 7. Translation to practice | ◍ | | | ● |

Note: The full black dots ● indicate a main source of information for the particular step, while the full grey dots ◍ indicate only a minor influence and guidance for the particular step.

participants were asked open-ended questions about the practical implementation as well as barriers and facilitators of the component. Lastly, results were summarised and participants were asked for final feedback. Table 1 provides an overview of the different methodologies used in each development step.

### Data analysis

We used data triangulation to verify the results. Our main source of information was the working group sessions, as they provided the richest data on the perspectives of the different target groups. Individual interviews, observations and literature research were used to complement and check the information obtained throughout the working group sessions. Working group sessions and individual interviews were recorded and transcribed non-verbatim. A thematic analysis was conducted based on the steps described by Braun and Clarke.[44] Common themes were identified within each step of the development process (eg, lack of sufficient communication). Afterwards, the themes from all sessions were compared and associations were found between, for example, identified problems and determined needs, or current scientific evidence and best practices of the organisation. The results of each session provided input and structure for the next session. After each working group session or individual interview, a member check was conducted by summarising the most important findings. The results were discussed regularly within the research team. The research team consists of all authors involved. They work in the field of social gerontology, public health and primary and long-term care. A research logbook was kept by the researchers to keep track of all research activities and intermediate results, together with field notes from these activities.

### RESULTS

The following section presents the final model of the I-MANAGE programme in detail, starting with the developed logic model, and eventually the translation to practice, which contains a detailed description of all programme components. A detailed description of the results from the first five programme development steps is provided in online supplemental file 3.

### Development of a logic model

The logic model starts with the aim and subaims of the programme based on identified problems and needs (*result from development steps 1 and 4*), which are derived mostly from the working group sessions and individual interviews. To fulfil these aims, six programme components were determined: (1) improving assessment and goal-setting; (2) stimulating self-management during meaningful daily activities; (3) optimising the use of the physical environment; (4) optimising the use of the social environment; (5) improving interprofessional collaboration; and (6) supporting the informal caregiver. These originate from both the input from stakeholders (*result from development steps 2, 4, 5 and 6*) and evidence-based practices from the literature (*result from development step 2*). The intended client outcomes of the programme are reducing (I)ADL disability, improving self-management skills, increasing QoL of both the client and informal caregiver and reducing healthcare usage and expenditures (proximal outcomes), which are common outcome measures in reablement programmes abroad. Eventually, improving these proximal outcomes would help the older adults to remain living at home independently and avoid unnecessary transitions to institutional care (distal outcomes). Figure 2 presents the logic model.

### Translation to practice: description of the I-MANAGE programme

The I-MANAGE programme, as described here, is the result of input from stakeholders and literature research during the last step of the development process (step 7).[41] The following sections describe the programme in detail. The programme, as described, may serve as a model for the use in different local settings and leaves room for tailoring to the specific needs and resources of the organisation.

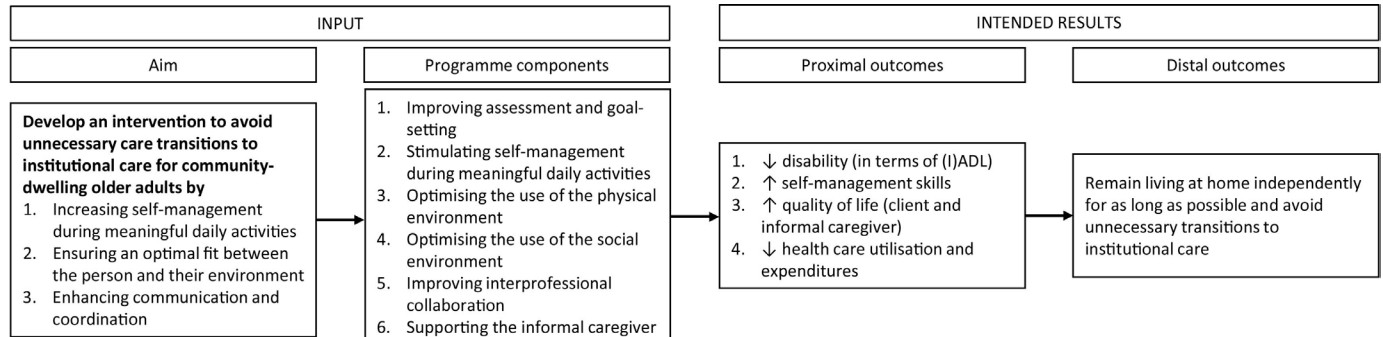

**Figure 2** Logic model of the I-MANAGE programme. The logic model of the I-MANAGE programme presents the aim and associated subaims of the programme, the programme components, and proximal and distal outcomes. (I)ADL, (instrumental) activities of daily living.

To achieve the intended outcomes presented in the logic model, the programme should intend to apply a two-tiered approach: (1) focusing on the capabilities of the client and (2) paying attention to environmental factors (ie, physical, social and organisational). I-MANAGE should be intended for community-dwelling older adults who are still able to (re)learn new skills and apply them in daily life without an indication of terminal illness or planned nursing home admission. The average duration of the programme is preferably 8 weeks according to the members of the working groups.

### Interdisciplinary collaboration
I-MANAGE aims to facilitate interdisciplinary collaboration by encouraging intensive collaboration between the reablement team, client, informal caregiver and other care professionals delivering care and support to the client and informal caregiver. The programme should be delivered by an interdisciplinary reablement team, generally consisting of an RN, an OT and, depending on the local context, supplemented with a PT, social worker, DSW or other disciplines. The reablement team should initiate the programme and is responsible for the direct support of the client and informal caregiver. A care coordinator should be appointed, which in most cases, is the OT. Depending on the necessary care and goals set by the client, the reablement team may consult other care professionals (eg, usual home care team, psychologist or general practitioner); for example, during the intake phase for advice or while working on the client's goals. They should be informed and coached by the reablement team.

Furthermore, regular team meetings should be organised (eg, (bi-)weekly) to discuss the intake of new clients and informal caregivers, the progress made by clients and the final evaluation of clients' personal goals. Additionally, team meetings could provide the opportunity to discuss specific cases with other team members and to organise training. All members of the reablement team should have access to access the care plan, report progress and follow-up on each other's work, preferably through a shared electronic care file (ECF) of the client. Lastly, when discharged from another care facility (eg, hospital or rehabilitation facility), a smooth handover of client information should be initiated so no necessary details of the client's care process are lost.

### Practice-oriented training
Preceding the programme, the reablement team should receive practice-oriented training focusing on adequately delivering I-MANAGE. Within this training, care professionals should receive a detailed manual beforehand, consisting of background information and a description of the programme, including the goal, key components and I-MANAGE's care process. The manual was developed by the researchers and revised by working group members to make sure it is suitable for practice. Additionally, a toolkit for achieving individual client goals should be provided (eg, exercise booklet based on the OTAGO exercise programme (a programme aimed at reducing fall incident in older people),[45] social map including welfare initiatives and tools to assess needs and wishes of the caregiver). The training should consist of multiple sessions, preferable in the following structure; the first is a kick-off meeting for the reablement team, focusing on the goal and content of I-MANAGE. Second, specific training sessions occur for OT and PT concerning the use of the Canadian Occupational Performance Measure (COPM)[46] to set personal goals with the client and the OTAGO exercise booklet, respectively. Finally, the reablement team receives a booster session to practice and discuss challenging situations and focus on motivational and conversation skills. The kick-off and booster sessions each last 2 hours and the specific sessions for OT and PT last 1 hour. Additionally, the programme should stimulate training on the job, meaning that members of the reablement team can coach each other and external care professionals based on their own expertise.

### I-MANAGE care process
The I-MANAGE care process consists of five consecutive phases: (1) initiation; (2) intake; (3) care plan; (4) care delivery; and (5) evaluation. Each phase is described in detail below and presented in figure 3. The five phases are a result of the practical translation of the previously described logic model.

**Referral of the client to I-MANAGE programme:**
- Community nurses
- General practitioner
- Institutional care facilities

**Start of the programme**

**Phase 1: Initiation**
1. Provide information to client (and informal caregiver)
2. First visit by OT
   - Exploratory conversation based on Positive Health
3. If discharged from other care facility; initiate smooth handover

**Week 1**

**Phase 2: Intake**
1. Environmental assessment by OT
2. Set meaningful goals by using COPM
   - Guided by exploratory conversation from phase 1
3. Intake with informal caregiver to assess burden and needs

**Week 2**

**Phase 3: Care plan**
1. Determine interventions and actions to reach goals
   - Guided by preferences of client/informal caregiver
   - Specific attention for client's capabilities, social network and physical environment
2. Recorded in electronic care file

**Week 3-8**

**Phase 4: Care delivery**
1. Care provided as described in care plan
2. Follow-up on the care process by reablement team
3. If needed, initiate additional support for informal caregiver

**Week 8**

**Phase 5: Evaluation**
1. Continuous evaluation during care visit and team meetings
   - Continue, adjust or terminate care plan
2. Formal evaluation after 8 weeks by using COPM

**Aftercare**
1. Continue programme for maximum of 2 weeks
2. Referral to usual care

**Interdisciplinary collaboration:**
- Appointed care coordinator
- Regular team meetings
- Shared electronic care file
- Coaching on the job

**Figure 3** Detailed presentation of the I-MANAGE care process. I-MANAGE has a 5-phase care process (initiation, intake, care plan, care delivery and evaluation), preceded with the referral of the client through different routes. After an average duration of 8 weeks, clients are referred to aftercare. Interdisciplinary collaboration is a continuous element of the programme. COPM, Canadian Occupational Performance Measure; OT, occupation therapist.

### Phase 1: initiation

Programme referrals could be done through community nurses, general practitioners or institutional care facilities. Community-dwelling older adults eligible for I-MANAGE should receive information about the programme. The care coordinator has to plan a first visit to present the programme to the client and, if applicable, their informal caregiver to provide the opportunity to ask additional questions. The care coordinator should initiate an exploratory conversation based on the principles of positive health to gain insight into the client's needs and wishes.[47]

### Phase 2: intake

Within the first week after the initiation phase, the OT should perform an environmental assessment, identifying necessary home modifications and assistive devices to ensure a safe environment. The environmental assessment is not limited to the inside environment but does also include the entrance and outside environment. Additionally, the care coordinator, or an assigned social worker, should perform an intake with the informal caregiver assessing their needs and wishes. Furthermore, the OT must set meaningful goals with the client using COPM[46] (maximum of five goals). This instrument requires the clients to score both their performance and satisfaction when performing these activities.[46] Goal setting is guided by the exploratory conversation held in phase 1. The ultimate goal is to improve the client's participation and well-being; therefore, goals should not merely be (I)ADL-related (eg, meeting friends at the local café, or painting in the hobby room on the first floor). Goals should be recorded in the client's ECF.

### Phase 3: care plan

Possible interventions and actions to achieve the client's goals and provide the informal caregiver with the right support should be discussed within the reablement team. These interventions and actions are derived from the available toolkit. Afterwards, the care coordinator should determine the final interventions and actions with the client and informal caregiver, guided by their preferences and capabilities, possibilities of the social network and the physical environment. Subsequently, the care plan should be recorded in the client's ECF and shared with all members of the reablement team. The intensity of the programme depends on the care needs of the client and their preset goals, which may require a higher intensity in the beginning but less at the end when (sub-)goals are (partly) reached.

### Phase 4: care delivery

The reablement team should deliver care and support as described in the care plan. Care delivery should be coordinated by the OT, who is also responsible for assisting with (re)learning and practicing meaningful daily activities. The RN is responsible for supporting managing the client's personal care needs. The PT, when part of the reablement team, is responsible for functional training and stimulating participation in daily life. The client's environment should be adapted (eg, home modifications, assistive devices and care technology), to support them in (re)learning activities. While supporting meaningful activities, the client's self-management should be stimulated by practicing activities and ensuring that tasks are not taken over by care professionals or informal caregivers. If necessary, a social worker should provide the informal caregiver with additional support to decrease their burden (eg, respite care).

### Phase 5: evaluation

The evaluation of the care process should be structurally embedded in interdisciplinary team meetings. Based on conclusions drawn during these meetings, the care coordinator can decide whether the care plan is continued, adjusted or terminated. This must be discussed with the client and informal caregiver. At the end of the programme, a formal evaluation of the client's goals should take place using COPM,[46] including scoring the performance and satisfaction within activities. Afterwards, the care coordinator decides with the client and informal caregiver if the programme should be extended (maximum 2 weeks) to ensure all goals are sufficiently reached or if the client needs referral to usual care.

## DISCUSSION

This manuscript describes the development and content of I-MANAGE, a community-based reablement programme. The programme was developed for community-dwelling older adults to improve their self-management and participation in daily life and ensure that they can remain living at home independently as long as possible and avoid unnecessary transitions to institutional care, while also increasing QoL and decreasing informal caregiver burden.

The programme contains several key elements that are considered essential and should be present when implementing the programme in any care setting. First, in line with the conceptual definition of reablement,[18] interdisciplinary collaboration is important in I-MANAGE. However, how this element is implemented in practice depends on the contextual circumstances of a country or region and the resources available. In this study it was recommended by the Dutch stakeholder to appoint a care coordinator, schedule (bi-)weekly meetings and implement a shared ECF. Recent literature indicates that investing in interdisciplinary collaboration stimulates patient-centred care by ensuring a holistic view of the client's situation and creates shared responsibility.[48–50] This is achieved by good communication and coordination within the team but also with the client and informal caregiver.[51] Moreover, delivery of the programme by an interdisciplinary team, including allied health, such as OTs, is deemed valuable because of their educational background.[30 52] It also stimulates continuous learning and is experienced as exciting

and constructive by care professionals.[50 51] Additionally, the integrated practice-oriented and on the job training, where care professionals can learn from other disciplines, help to invest in the self-efficacy of care professionals. This is essential, because successfully changing behaviours remains a challenge. The training entails several key topics as mentioned before, however, depending on the local context the extent of the training may vary, for example, due to previously received education or training. Second, at the start of the programme (phase 1 and 2), we implemented a standardised goal-setting tool, COPM,[46] preceded by an extensive intake based on the principles of positive health.[47] Previous research has indicated that using standardised assessment or goal-setting tools could increase the effectiveness of reablement interventions, and is therefore considered an essential element of the programme.[30] Additionally, it increases client involvement and helps professionals to identify meaningful activities with the client.[53] Third, when delivering care (phase 4), the programme integrates several important aspects. First, supporting informal caregivers is assumed to contribute to the effectiveness of I-MANAGE. Previous research found that providing informal caregivers with the right psychosocial and educational support strengthens their ability to cope with their new role.[54 55] This is an important addition to most reablement programmes because, although making use of the clients' social network is a strategy to reach their goals according to the internationally accepted definition,[18] this is often overlooked.[56 57] Additionally, I-MANAGE promotes optimal use of the social and physical environment, which is essential since a demanding environment can either stimulate or hinder a person's participation in meaningful activities.[3 58] Lastly, the programme stimulates self-management through participation in meaningful daily activities, which is a core element within reablement ('doing with…' rather than 'doing for…' the client).[18 59]

The described programme serves as a model and leaves room to tailor the intervention to a specific context and the needs of the organisation. For example, the programme leaves room for the organisation to choose which target group would benefit most. Additionally, delivering the programme by an interdisciplinary team is a prerequisite. However, depending on available resources, the organisation can decide on the composition of the reablement team. Moreover, I-MANAGE is a multicomponent programme in which organisations can integrate their own innovative practices, especially when working towards and reaching clients' meaningful goals. Also, the duration and intensity of the programme may vary according to the needs of the local context and chosen target population. Moe and Brinchmann[25] confirmed the necessity of tailoring reablement services to local conditions by arguing that establishing reablement in an existing organisational structure is a complex process. Apart from available resources, during the co-creation process, some influencing factors for implementation were identified. For example, different information

technology (IT) systems, which make communication and information transfer less evident. Moe and Brinchmann[25] mentioned that, next to communication and IT systems, habitual ways of offering health services, a lack of knowledge about the rehabilitation potential of older adults and active ageing benefits are also experienced as barriers. A recent scoping review identified several factors that act as barriers and facilitators during the implementation of care innovations, among which, available resources and communication were listed as possible barriers to implementation.[60] It is critical to consider these factors when implementing I-MANAGE in practice. In order to overcome several barriers to the implementation of I-MANAGE, it could be useful to set up advisory boards and working groups to discuss the implementation process and adjust where needed. Additionally, investing in getting the management of care organisations on board to support the implementation within their organisation and setting up knowledge exchange between sites where reablement is being implemented to share experiences and best practices could be beneficial. Investing in a suitable organisational structure is essential because it remains a challenge to successfully change existing organisational structures.[25 61 62] Regarding future research, further knowledge is needed to explore feasibility and (cost-)effectiveness of I-MANAGE, as it has not yet been proven. Since the programme is very context specific and can be tailored according to the needs and resources of an organisation, it would be beneficial to investigate what works for which target group and under what circumstances, for example, by means of a realist evaluation.[63] The programme is currently being implemented and evaluated at different Dutch care providers. The systematic reporting of the development of the programme provides useful insight for future research looking to develop complex health interventions or to implement a similar programme.

A strength of I-MANAGE and its development is that it is both theory and evidence-based,[42 43] which has proven to be advantageous when developing effective interventions.[64 65] Moreover, Thuesen et al[66] highlight the demand for making theory explicit in reablement interventions and in addressing the physical, psychological and sociocultural perspectives of ageing within these interventions. Another strength of the development process was the co-creation process, which gave a voice to multiple stakeholders and made the programme suitable for practice. We used the development approach described by Bleijenberg et al,[41] which combined a range of published approaches to intervention development to enrich the MRC framework. This approach was chosen because using the MRC framework further assists us in evaluating and adapting the programme. However, we are aware that multiple approaches to intervention development exist as described by O'Cathain et al.[67] These different approaches share many similarities (eg, stepwise approach or involvement of stakeholders), but there are also significant differences (eg, the focus on implementation or

theory). It is important to acknowledge these differences and always choose an approach best suited to the purpose of the research. Additionally, most of these approaches are not set in stone and leave room for the researcher's own interpretation. It must also fit the setting and timing of the development process. There are also some limitations related to this co-creation process. First, during the process, we always informed management to obtain their support, but they were not included in the working groups, which could have blocked important insight. Second, we aimed to include all end users of the programme during the development process (care professionals, clients and informal caregivers). However, we only included client and informal caregiver representatives in the working groups and interviewed only two informal caregivers between sessions. Additionally, care professionals were recruited from the professional network of researchers and were recruited in a convenient way. We therefore did not ensure variation among these participants in terms of, for example, gender, age or years of experience. Lastly, we obtained data from many different sources (ie, scientific literature, individual interviews, working groups and observations), which made it difficult to find common ground throughout all the sources and forced compromise. However, this use of data triangulation is also a strength of the development of the programme as findings could be checked multiple times with different sources, increasing the validity of the results.

**Author affiliations**

[1]Department of Health Services Research, Faculty of Health Medicine and Life Sciences, CAPHRI Care and Public Health Research Institute, Maastricht University, Maastricht, The Netherlands

[2]Living Lab in Ageing and Long-Term Care, Maastricht, The Netherlands

[3]Department of Public Health and Primary Care, Academic Centre for General Practice, KU Leuven, Leuven, Belgium

[4]Department of Public Health and Primary Care, Academic Centre for Nursing and Midwifery, KU Leuven, Leuven, Belgium

[5]Faculty of Medicine and Life Sciences, UHasselt, Hasselt, Belgium

**Acknowledgements** We would like to thank the Dutch stakeholders (from MeanderGroep Zuid-Limburg, Vivium Zorggroep, Maastricht University, Cicero Zorggroep, gemeente Kerkrade, and Burgerkracht Limburg) who have participated in the working group sessions, observations and individual interviews, and therefore have helped with the development of a reablement programme suitable for the Dutch home care setting.

**Contributors** All authors—IM, HV, GIJMK, JCMvH, EV, GG, SFM—were involved in the development of the I-MANAGE programme. IM wrote the first draft of the article. All authors reviewed the manuscript and approved the final version. IM and SFM act as guarantor for the manuscript.

**Funding** This work was supported by the European Union's Horizon 2020 research and innovation programme under the Marie Skłodowska-Curie grant agreement No 812656 as part of the TRANS-SENIOR project (www.trans-senior.eu).

**Competing interests** None declared.

**Patient and public involvement** Patients and/or the public were involved in the design, or conduct, or reporting, or dissemination plans of this research. Refer to the Methods section for further details.

**Patient consent for publication** Not applicable.

**Ethics approval** As participants were not being subjected to actions and no rules of behaviour were imposed on them no ethical approval was needed according to the Dutch 'Medical Research Involving Human Subjects Act' (WMO). Nevertheless the study was carried out in accordance with the principles of ethical research and all participants involved in the working group sessions and interviews provided either written or verbal informed consent prior to the start of each session. Participants gave informed consent to participate in the study before taking part.

**Provenance and peer review** Not commissioned; externally peer reviewed.

**Data availability statement** Data are available upon reasonable request. The data sets used and/or analysed during the current study available from the corresponding author on reasonable request.

**ORCID iD**

Ines Mouchaers http://orcid.org/0000-0001-7189-4468

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
