## [Reviewer comments · BMJ Open]

ARTICLE DETAILS

TITLE (PROVISIONAL)	Development and content of a community-based reablement programme (I-MANAGE): A co-creation study
AUTHORS	Mouchaers, Ines; Verbeek, Hilde; Kempen, Gertrudis; van Haastregt, Jolanda; Vlaeyen, Ellen; Goderis, Geert; Metzelthin, Silke

VERSION 1 – REVIEW

REVIEWER	Marianne Eliassen UiT The Arctic University of Norway Faculty of Health Sciences
REVIEW RETURNED	02-Jan-2023

GENERAL COMMENTS	Dear Authors, Thank you for producing research on such an interesting subject. I have done my best to make comments that I believe can contribute to an improved product, and hope it finds you well. I think it is exiting that you have used a co-creation methodology to co-create a model that emphasize stakeholders views. I also agree that reablement practices need a more solid foundation (both scientific and theoretical), which I believe this research can contribute to. However, I do have some remarks that could make the paper clearer. Title: The title indicate that the article gives insight about both the development, the rationale, and the content of I-MANAGE. In a way this is what you attempt to, however, I believe that this is a very wide and ambitious, and only manage to touch the aspects briefly. However, I believe that the article has the potential to provide a solid overview of a model for preventing I-MANAGE. While a thorough depth of the rationale of the model lacks (to my opinion). Perhaps you would benefit from scaling down the ambition of the title (and aim)? Abstract: For readers that do not know about this work it is a bit confusing what you write about the TRANS-SENIOR network. Was the aim to aid implementation of reablement, or describe a model for reablement? The aim is not clear. You state that this paper provides a 'blueprint'. Are you sure you want to use this word? A blueprint is a tool to make a solid copy that is exact the same as the original. If a central aspect was that the model should be flexible and adaptable to contextual variations, maybe a 'blueprint' is not the right word? How about "model" or "framework"? Is there missing an 'in' at p. 3, line: 30? "The interviews and working group sessions took place IN Dutch [...]" Background: This section is well structured and clear. However, I miss some information about what has been done before, for example: what
---

	contexts/countries have implemented reablement interventions? How is implementation of reablement described in the literature? What barriers and facilitators are identified in research? Why do we need the description that you provide in this article (we do, but convince the reader). How does this study differ from other studies that describe reablement processes? Can you highlight what new knowledge you can add to the field through this study? In the background, you state that the development of I-MANAGE is suitable for the Dutch community care setting. Can you provide more background about this setting? It is not very clear how you have been working with the contextual adjustments in the process throughout the article. Methods: You describe that the project is a co-creation between researchers and stakeholders. Who initiated the project? Researchers? Governments? Patient organizations? Care providers? (Whose needs are the project targeting? And who “owned” the co-creation process, and how does the ownership affect the collaboration and participation in such co-creation process?) You state that end-users and stakeholders were involved in the co-creation process, however, they were not involved in the study design-process or dissemination of the findings. Is my assumption about this right if I say that the reablement model was co-created, but the research was not? I think this is an important distinction that you should provide clarity about. I would like you to provide more information about the included stakeholders: How many participated in total? From where were they recruited? Did they know about each other beforehand? Did they work together? (inter-relationships may affect the co-creation process, how was this a case in your study?) Will a table of the different stakeholders, and the varied activities they participated in give a better insight? At p. 8, line 15 you state “The logic model systematically visualizes...”. The reader is not provided with any background information about a logic model concept, and this may be confusing. Can you provide more information about this? You state that you have done a literature review and refer to an earlier publication. Can you provide a brief overview of what was done, and what you found that became relevant for this model, and how it was dealt with? You write that you conducted three sessions. How long time was it between the sessions? What happened in between the sessions? Were any of the initiatives tried out in this period, or are the results only based on discussions about vision and ideology? It is a bit confusing to read p. 8 where you write about the six steps of developing the model, that is not chronological due to the varied methods you conducted (Did you conduct step 2 and 3 (literature review) first, then step 1? It is not very clear). Please rephrase in a way that provides clarity. (A figure that display the chronology of the process may help) You also state that three geriatric experts participated. What kind of experts are they? What profession? From what work context are they recruited? Did they have experience with reablement? There is not enough information about how the observations were conducted. Was this carried out at the users’ home? How many of the stakeholders conducted the observations? Was it only done by the researchers? What was the aim of the observations? What was in focus? What data did this generate? Video material? Field notes? Informal interview notes? What was the context they were observing? Were they already trying the I-MANAG intervention (or
--	---

	any other form of reablement)? Or were the observations done in traditional health care settings? At p. 9, line 26 you state “[...] resulting in three to four members per working group” How many working groups were conducted? Did they focus on the same themes? Did you use any interview guide? Thematic template or guide? What was the focus in these groups, how was it facilitated and how were they conducted? Were the user representatives and informal caregivers in mixed groups with other stakeholders? If so, did you do any interventions to equalize the power relationship between the involved actors? The last sentence in the methods section is “The results were discussed regularly within the research team” Who is the research team? Please provide some sentences about research reflexivity. Results: A general objection is that the results are written in a way that does not make it visible that they are the empirical result of the study, which makes me question what you base your statements on. For example p. 12, line 21: “the program is delivered by an interdisciplinary team, generally consisting of an RN, an OT, and [...]” Is this results based on the study? Is it based on the literature review or the co-creation activities? Or is it just the authors assumption of general practice? This is not clear, and there are several examples of this throughout the results section. I think a rephrasing may help you with clarifying this. It is however important that the results you provide are actual results of this current study. More examples: “A care coordinator is appointed, in most cases, the OT [...]”: Based on what?, “The reablement team may consult other care professionals” Based on what? (and how should that be done?), “(bi-) weekly team meetings are organized to [...]”: Based on what? At p. 11, line 23-24 you write: “The intended results are reduced (I)ADL, [...]” I guess you mean that “the intended patient outcomes of reablement are [...]” This needs rephrasing so that readers do not get confused about the results of the study and the results/outcomes of the interventions. P. 11: “Development of a logic model”: This section is a bit confusing to read as it describes six stages of the method, three sessions conducted, six program components, and five phases of I-MANAGE. Can you find any ways to just make this easier for the reader to follow? At p. 13 you describe a practice-oriented training for the reablement staff. Who did/should provide this training? And how was the ‘detailed manual’ developed (and by whom?) and can the manual be made available for the reader in any way (if this is supposed to work as a ‘blueprint’ it should be made available). Was the content of this training program also a result of the co-creation process, or literature review? Again, this is a bit unclear. You have based the initiatives on some standardized tools, such a COPM and OTAGO. What was the rationale behind this? Is it based on results from the co-creation process? Literature review? Or just a random choice? Presentation of the I-MANAGE process: Again, I am having problems with reading this text as a presentation of study results. For example, the sentence at line 53; “People are referred to the program through community nurses [...]” What about writing: “a common perception in multi stakeholder discussion, was that referrals should be made by the community nurses”, or “based on the literature review, it became clear that a referral from the community nurses is advantageous”. Try to show the reader what evidence that support your claim.
--	---

	Page 14, line 3-6: You claim that "The care coordinators initiates an exploratory conversation [...]", and then you refer to reference number 37. Are these results based on the literature review? Again, it is not clear what the evidence are. At p. 14, line 39 you refer to an "available toolkit". What toolkit is this? Can it be made available for the reader? Discussion: You state that the model leaves room for contextual adjustments. I miss a description of how this adjustment was done in this current study, concerning the Dutch context. You also state that an example of adjustment can be to utilize other professions or care staff. Does it matter what professions that provide the model? What was the rationale behind utilizing OTs and RN in this study? At p. 18, line 39-43 you describe results from a scoping review, and refer to the review, but also an additional study, which is not the review. Why is this? I apologize for probably making you work some more with this article. I do believe this is an interesting study, and I also think it is important that you have included relevant stakeholders in this process. However, I think the abovementioned should be improved, so that others can benefit from this work. Thank you so much for letting me read this. I look forward to reading an improved edition.
--	---

REVIEWER	Magnus Zingmark Municipality of Östersund, Health and Social Care Administration
REVIEW RETURNED	09-Jan-2023

GENERAL COMMENTS	Dear authors, It was a pleasure to read your manuscript which I hope will soon be published after some revisions. It is an important contribution to the field of reablement based on an ambitious co-design approach. However, as seen in my comments, I suggest several revisions which I think could enhance clarity of the paper. I look forward to read a revised version of the manuscript. Background A general comment that you might consider in the first sentences is that the language can be interpreted as slightly ageist. While disability becomes more frequent as people age, a large proportion in the older population in fact remain independent and can live an active life. For people beginning to experience disability, it is often minor disability from the beginning. As the text is written now, the reader can get the impression that most older people will suffer from severe disability. See for example Gore or Zingmark that disability rather can be emerging over time. https://academic.oup.com/ageing/article/47/6/764/5079486 https://bmcgeriatr.biomedcentral.com/articles/10.1186/s12877-021-02283-x In the end of the first paragraph: "Not tackling these challenges properly could lead to increased healthcare utilization and costs [13], and unnecessary (re)hospitalisations or permanent nursing home placement..." I think it also could be increased use of social care services (it would be home care in a Swedish context). Please adjust if this applies to the context of your country. In general, I agree with your statement that "reablement programmes,
---

are often insufficiently described in the scientific literature. But not always, see for example these references:
<https://www.tandfonline.com/doi/full/10.1080/11038128.2022.2089229>
<https://bmccgeriatr.biomedcentral.com/articles/10.1186/s12877-022-03185-2>
<https://onlinelibrary.wiley.com/doi/abs/10.1111/hsc.12934>

Adding your paper, once published, the literature is becoming increasingly better to describe the actual content of reablement and how it is delivered.

The last paragraph is quite long which lead to that the study objective is not 100% clear. Could some information go into the methods and even discussion (“Due to its extensive description, this manuscript may serve as a blueprint for implementation in other local settings?”).

Methods

Patient and public involvement

Who were “All end-users”?

Were it those who were to implement the program or the target population who needs were to be addressed. Or both?

Setting and participants

You write: “The programme was developed for community-dwelling older people who wish to remain living at home independently”. First, doesn’t almost all older people want to do that? In that case, did you have more specific criteria?

In this paragraph, it is not clear how participants were recruited and informed about the study (check how this match with your section about ethics).

There were working groups. How many? Who were included in which working group? How often did they meet and over how long time period? Or was it only one working group?

As I read, I understand that the program was developed in one local context. Could you provide some additional information about this city or municipality, population size, was it similar to other contexts in your country? Since you later argue that a programme like I-MANAGE can be tailored to local contexts, here could be a place to provide a good basis for this line of reasoning.

I agree that ethics were not required for this study. However, you do not have any information about participant characteristics which is a limitation. Did you do any efforts to ensure variation among those recruited e.g., in terms of age and gender, experience of reablement... Please write something about this and if not anything was done, name it as a limitation.

Development of a logic model (1-6)

Who led the working groups?

There were rather large groups (9 participants + researcher). Did you do anything to ensure that all participants had the chance to be involved in the co-design process?

Was the group size a limitation?

You mention the steps, but it is confusing that you do not mention them in order 1, 2, 3,...

In addition, it is also unclear when you performed this: “Lastly, observations were performed to examine current practice and context of home care services (step 5) and to identify problems (step 1). Six observations of half a day each were performed with a DSW, an RN, and a nursing assistant, and three days with allied health.

The first step is to identify the problem. I understand that “the problem” is what the program will address but could you explain that so it is VERY clear.

You do a lot to get an accurate picture of the problem, right? However, the logic presentation of what you have done and when you did it isn’t completely clear. I suggest something in line with: To ensure a valid identification of the problem we conducted observations, individual interviews discussions in the working groups...., All subsequent steps was based on the identified problem....

Was it the same people that were included in the 11 interviews? If yes, please clarify. If no, please describe how they were recruited.

Data analysis

Did you also included data from ongoing communication: “Between sessions, working group members were consulted for additional input”

Step 7

You write: “For example, on informal caregiver support, we invited an informal care consultant, an informal caregiver representative, and a psychologist; resulting in three to four members per working group.” It is difficult to understand how many working groups there were. Can you please specify that? And again, who led the working groups, the same person or different persons?

Results

Clearly stating what the problem is. I think the text would benefit much if you should start with presenting what the identified problems were. Then, it will be much easier to understand why you decided to include the six programme components. It is clear in supplementary file 2 but I think the reader should access this information much more easily since it is so central for your results.

Further, in supplementary file 2, you say that “There appeared a gap between the person’s abilities, needs, and wishes, and the environment they reside in. When older people experience a deterioration in health status, their environment is often not adapted, i.e. the necessary home modifications and assistive devices are not always in place”

You refer to Wahl et al from 2009. First, I think the reference is a bit old, is there something more up-to date? Second, I wonder if Wahl et al., refer to a similar population as for which your programme was developed? Please consider carefully is this reference is appropriate.

I miss a clear structure that follows the logic of the model. I think that the logic can provide an easy structure that can help the reader understand step by step what you found. Please consider to use the logic and the order 1-7 not only in the supplementary file.

	Step 1 identifying the problem Step 2 previously identified evidence Step 3 theoretical foundation for the programme Step 4 determination of needs Step 5 examination of the current practice and context Step 6 modelling the process and outcomes Step 7 translate components into practice Description of the programme There is a discrepancy which I think could be made clearer. You write: “I-MANAGE is intended for community-dwelling older people” and later “I-MANAGE aims to improve interdisciplinary collaboration”. Thus, is the programme designed to improved delivery of reablement? That is, is it designed to help professionals to initiate and deliver the service, which in turn benefits the clients? I wonder why I-MANAGE is intended for community-dwelling older people? Isn’t reablement a service intended for clients irrespective of age (see definition based on Delphi study) Why is the duration 8 weeks? Was this a result of the co-design process? Training OTAGO is included and usually this is implemented to reduce falls. However, it is not clear from the identification of problems, determining needs or modelling outcomes that fall-related injuries is something that needs to be more specifically addressed in reablement. Is that information missing? Or can you elaborate the text why OTAGO needed to be included? In addition, it is not clear if and how OTAGO is implemented when delivering the programme (phase 4). Also in figure 1, any text on OTAGO is missing. “The training consists of multiple sessions”. Can you be more specific and write exactly how many sessions were included and over how long time training was implemented. After how long time was the booster session? Phase 5 “The evaluation of the care process is structurally embedded in interdisciplinary team meetings”. How often are these meetings? Discussion When reading the discussion, I get the impression it could be elaborated further. The authors state “The described programme serves as a blueprint and leaves room to tailor the intervention to a specific context”. Although I agree, I do not think the results or methods emphasized adaptation to the local context sufficiently to raise highlight this early in the discussion. I rather miss a distinct discussion of the findings presented in this study compared to previous research. Also here, the logic of the model could provide a structure to reflect over the results and the methods used. For each of the following, to what extent was your findings similar or different to previous research? Step 1 identifying the problem Step 2 previously identified evidence
--	--

	Step 3 theoretical foundation for the programme Step 4 determination of needs Step 5 examination of the current practice and context Step 6 modelling the process and outcomes Step 7 translate components into practice Another or an additional perspective could be to discuss the five phases in the I-MANAGE care process; to what extent was your findings similar or different to previous research? Further, as described in the methods, steps 1-6 were about development of the programme whereas step 7 was about translating and choosing relevant actions for optimizing delivery of the programme. What was the pros and cons of this approach? Were there similarities or differences when comparing to other programmes that were based on co-creation. See for example https://www.tandfonline.com/doi/full/10.1080/11038128.2022.2089229 in which there are some similarities but also clear differences in (a) the scope of stakeholders involved in the co-design process and (b) the scope of the programme. However, there was similarities in the phases described. In addition, different tenses (past, present) are used. References There are several references that are a bit old (10 years or more) and therefore you may consider to update some references, see suggestions in previous comments. Figures They are not numbered. Supplementary files Do the page numbers referred in the table match the actual page number in the document? I find it hard to follow.
--	---

VERSION 1 – AUTHOR RESPONSE

Response to concerns raised by reviewer 1	
1. I think it is exiting that you have used a co-creation methodology to co-create a model that emphasize stakeholders' views. I also agree that reablement practices need a more solid foundation (both scientific and theoretical), which I believe this research can contribute to. Our response: We thank the reviewer for the compliments.	N/A
Title	
2. The title indicate that the article gives insight about both the development, the rationale, and the content of I-MANAGE. In a way this is what you attempt to, however, I believe that this is a very wide and ambitious, and only manage to touch the aspects briefly. However, I believe that the article has the potential to provide a solid overview of a model for preventing I-MANAGE. While a thorough depth of the rationale of the model lacks (to my opinion). Perhaps you would benefit from scaling down the ambition of the title (and aim)?	Title page p 3, p 8, p 17 and p 20

Our response: We agree that the rationale is not extensively described throughout the article. Therefore, we followed the advice and deleted the rationale from both the title and aim of the article.	
Abstract	
3. For readers that do not know about this work it is a bit confusing what you write about the TRANS-SENIOR network. Our response: We understand the confusion. However, it is a requirement of our funding agency to mention TRANS-SENIOR in the abstract and can therefore not be left out. To increase the readability we moved this information to the design instead of objectives. Explaining the network in detail is not feasibly with regard to the limited word limit.	Abstract p 3
4. Was the aim to aid implementation of reablement, or describe a model for reablement? The aim is not clear. Our response: We thank the reviewer for this question. The aim is to describe the development and content of a model for reablement, however, in the end this model can aid the use of reablement in multiple setting. We understand the formulation of the aim may be confusing and therefore adapted it: The aim of this paper is to describe the development and content of I-MANAGE, a model for a reablement programme for community-dwelling older adults. ... The I-MANAGE programme can be tailored to the local practices and resources and is therefore suitable for the use in different settings, nationally and internationally.	Abstract p 3
5. You state that this paper provides a 'blueprint'. Are you sure you want to use this word? A blueprint is a tool to make a solid copy that is exact the same as the original. If a central aspect was that the model should be flexible and adaptable to contextual variations, maybe a 'blueprint' is not the right word? How about "model" or "framework"? Our response: We agree with the reviewer that blueprint might not be the right wording. Therefore we changed 'blueprint' to 'model' throughout the manuscript.	Throughout manuscript
6. Is there missing an 'in' at p. 3, line: 30? "The interviews and working group	Abstract p 3

sessions took place IN Dutch [...]” Our response: We thank the reviewer for noticing this mistake. We corrected the sentence.	
Background	
7. This section is well structured and clear. However, I miss some information about what has been done before, for example: what contexts/ countries have implemented reablement interventions? How is implementation of reablement described in the literature? What barriers and facilitators are identified in research? Why do we need the description that you provide in this article (we do, but convince the reader). 8. How does this study differ from other studies that describe reablement processes? Can you highlight what new knowledge you can add to the field through this study? Our response: We thank the reviewer for these comments. We added the following parts to the background section to elaborate more on what has been done before, and explain the relevance of our study: [...] Because of its promising results, the interest in implementing reablement into everyday care is growing internationally. In Denmark, New Zealand and the United Kingdom reablement has been successfully implemented across more-or-less the whole country [1]. For example, in Denmark reablement in long-term care for older people is introduced by law since 2015, meaning that all municipalities must offer reablement interventions and all applicants for home care are assessed for potential for reablement, before being offered conventional home care [2]. Despite the promising results and successful implementation abroad, this is no indication that it is also effective in its current format in the Dutch home care setting due to contextual differences [26]. Implementing reablement is a complex process and if influenced by multiple factors, such as organisational factors, individual and social attitudes towards a new form of care, technological factors related to communication, and financial factors [27, 28]. It is crucial to critically consider these factors in the design, delivery, and evaluation of reablement [29]. Furthermore, to implement reablement in the Dutch home care setting, existing programmes need to be revised and adapted to suit the current context, which is crucial when developing and delivering complex health care interventions [29]. However, the development and content of community care programmes, such as reablement programmes, are often insufficiently described in the scientific literature [21, 30-32]. There are only limited articles available that describe either the development of the programme or its content in detail [33-35], and often these descriptions are included as part of a feasibility or pilot study [31]. This offers little guidance for replication of these programmes or to build on their previous findings, despite this being essential for the development of new and implementation of existing programmes in different settings [31, 32, 36].	P 7-8

Therefore, this paper describes the development, using a co-creation process, and content of I-MANAGE, a reablement programme for community-dwelling older adults to improve older persons' self-management and participation in daily life, while also increasing quality of life and decreasing informal caregivers' burden. By describing the development and content of the programme in detail, we increase the replicability and prevent other researchers from re-inventing the wheel. The programme is specifically suitable for the Dutch community care setting, however, due to its extensive description, this manuscript may serve as a model for implementation in other countries as well.	
9. In the background, you state that the development of I-MANAGE is suitable for the Dutch community care setting. Can you provide more background about this setting? It is not very clear how you have been working with the contextual adjustments in the process throughout the article. Our response: We thank the reviewer for this comment and understand that a description of the setting is lacking. We therefore included a description of the Dutch community-care setting in the methods section: I-MANAGE is based on international evidence and tailored to the Dutch long-term home care context. Home care in the Netherlands includes personal care (i.e. assistance with activities of daily living (ADL)), nursing care (i.e. medical assistance), and domestic support (i.e. assistance with instrumental activities of daily living (IADL)) [42]. Usually home care is funded by two statutory forms of insurance cover care: the Health Insurance Act (ZVW) and the Social Support Act (WMO). Clients often use a combination of ZVW (e.g. general practitioner care, therapists, hospital care, or medication) and WMO (e.g. domestic support, home adaptations) [42, 43]. Regarding the second comment about how we worked with the contextual adjustments, we would like to emphasize that reablement programs like I-MANAGE were not executed in the Netherlands, when the development of the programme took place. We used existing evidence from different countries to develop a programme suitable for the Dutch setting, together with different stakeholders from the Dutch community care setting. Their reflections on existing international programmes and knowledge about usual home care in the Netherlands led towards the contextual adjustments made.	P 10
Methods	
10. You describe that the project is a co-creation between researchers and stakeholders. Who initiated the project? Researchers? Governments? Patient organizations? Care providers? (Whose needs are the project targeting? And who "owned" the co-creation process, and how does the ownership affect the collaboration and participation in such co-creation process?) 11. You state that end-users and stakeholders were involved in the co-creation process, however, they were not involved in the study design-process or dissemination of the findings. Is my assumption about this right if I say that the	P 9-10

reablement model was co-created, but the research was not? I think this is an important distinction that you should provide clarity about. Our response: We thank the reviewer for these comments. We added some information about the co-creation process: The co-creation was initiated by the researchers as a response to the challenges that are caused by an ageing society (e.g., increasing care needs, decreasing staff). To deal with these challenges the Dutch government stimulates an ageing in place policy and promotes another way of (home) care delivery from “doing for...” towards “doing with...” clients or in the best case to enable clients to do things by themselves again. This developments are also adopted by Dutch care organisations, which hope to improve the quality and sustainability of their care services [41]. As mentioned under ‘patient and public involvement’, they were only involved in the development of the model, but not in the study design-process and dissemination of findings.	
12. I would like you to provide more information about the included stakeholders: How many participated in total? From where were they recruited? Did they know about each other beforehand? Did they work together? (inter-relationships may affect the co-creation process, how was this a case in your study?) Will a table of the different stakeholders, and the varied activities they participated in give a better insight? Our response: We thank the reviewer for this comment and agree that a table would be of added value. We therefore added Supplementary file 2 to the manuscript and the following text. The table was included as a supplementary file due to the journal’s guidelines for tables. The programme was developed in co-creation with stakeholders who participated in observations, individual interviews and working groups. Supplementary file 2 provides an overview of all stakeholders involved and the research activities they participated in. We also added a few sentences on the recruitment of the stakeholders: Participants were recruited from the professional network of the researchers. They were informed about the study and asked to participate via email. When	Supplementary file 2 and p 10

participants agreed to participate, verbal or written informed consent was provided before the start of each interview or working group session.	
13. At p. 8, line 15 you state “The logic model systematically visualizes...”. The reader is not provided with any background information about a logic model concept, and this may be confusing. Can you provide more information about this? Our response: We agree with the reviewer the lack of background information could be confusion. Some background was provided later on in the methods section (under Data collection), however it was limited and should be mentioned earlier in the methods section. Therefore we moved this section up, and added the following: First, a logic model was created. A logic model is a tool to illustrate how a programme will create change [39]. The logic model systematically visualises the aim and sub-aims of the programme, the programme components, and the intended outcomes [39].	P 9
14. You state that you have done a literature review and refer to an earlier publication. Can you provide a brief overview of what was done, and what you found that became relevant for this model, and how it was dealt with? Our response: We thank the reviewer for this comment. We choose to refer to the earlier publications for both literature reviews because describing the methodology again would not be possible within the word limit. Regarding the main results and conclusion of both publications, these can be found in Supplementary file 3, which describes the result of each programme development step (1 – 5).	Supplementary file 3
15. You write that you conducted three sessions. How long time was it between the sessions? What happened in between the sessions? Were any of the initiatives tried out in this period, or are the results only based on discussions about vision and ideology? Our response: We thank the reviewer for this comment. No initiatives were tried out in between sessions. Originally, we planned to pilot test elements of the programme in between sessions, however, due to Covid-19 this was not possible. The sessions only had a few weeks in between and the results of the first session were processed by the researchers to serve as the starting point for the next session, and so on. Each session focused on a specific step of the development process (as mentioned on page 12) and all 3 sessions had to take place before a concept version of the programme was created.	Figure 1, p 13 and p 11-12

However, we agree that a bit more information would be helpful to understand the process more. We added a sentence to provide clarity on what was done in between the sessions and, also in relation to other comments, created a figure (Figure 1) to provide clarity on the process:

Between sessions, working group members were consulted for additional input and clarification if needed. The researchers processed the results from each session and the additional information in order to be used as a starting point for the next session.

However, this information was also partly mentioned under data analysis. To avoid repetition, we deleted it here.

16. *It is a bit confusing to read p. 8 where you write about the six steps of developing the model, that is not chronological due to the varied methods you conducted (Did you conduct step 2 and 3 (literature review) first, then step 1? It is not very clear). Please rephrase in a way that provides clarity. (A figure that display the chronology of the process may help).*

Figure 1 p 13, p 11 and p 26

Our response:

We agree with the reviewer that it could be confusing for the reader to see how the process went. The model we based our development on (Bleijenberg et al) presents a non-linear and iterative development process. This, combined with the variety of methods we used, made it difficult to present it chronologically. However, we added a detailed figure (Figure 1) presenting the development process in more detail, and introduced the process a bit more in the beginning of the data collection paragraph:

Data collection was performed following a non-linear and iterative process as described by Bleijenberg et al. [40], and by using a variety of data collection methods including literature research, observations, individual interviews, and working groups. Figure 1 illustrated the development process in detail.

Caption of the figure:

Figure 1: The development process over time. This figure presents the development process over time from September 2019 until June 2021. The whole process is guided by scientific literature. The figure presents the two parts of the development process each with their related activities. The dark grey rectangles present the (intermediate) result of each part of the process. The number in between brackets represent the seven steps of the development phase as described by Bleijenberg et al. [40]. PT = physiotherapist; GR = geriatric rehabilitation; DSW = domestic support worker; RN = registered nurse.

17. You also state that three geriatric experts participated. What kind of experts are they? What profession? From what work context are they recruited? Did they have experience with reablement? Our response: We thank the reviewer for this comment. These experts were all academics who perform research in the field of geriatric rehabilitation. They did not have experience with reablement, as reablement was not very popular in the Netherlands yet when we developed the programme. We choose to include 'geriatric rehabilitation experts', as principles of reablement like goalsetting are applied in this setting as well. With regard to the word limit we did not added this information to our paper and only clarified which experts were involved: [...] and three experts in the field of geriatric rehabilitation research.	p 12
18. There is not enough information about how the observations were conducted. Was this carried out at the users' home? How many of the stakeholders conducted the observations? Was it only done by the researchers? What was the aim of the observations? What was in focus? What data did this generate? Video material? Field notes? Informal interview notes? What was the context they were observing? Were they already trying the I-MANAGE intervention (or any other form of reablement)? Or were the observations done in traditional health care settings? Our response: We agree with the reviewer that the description of the observations is quite limited. However, some of the asked information is already provided in the article (e.g. the aim of the observations and the focus). We also would like to point out that the number of observations was limited due to the Covid-19 pandemic. We added some information and adapted the paragraph: Lastly, observations were performed by the researchers. The observations were conducted in the traditional community care setting to examine the current practice and context of home care services (step 5) and to identify problems (step 1). Six observations of half a day each were performed with a DSW, an RN, and a nursing assistant, and three days at a geriatric rehabilitation facility. These observations resulted in field notes.	P 12
19. At p. 9, line 26 you state "[...] resulting in three to four members per working group" How many working groups were conducted? Did they focus on the same themes? Did you use any interview guide? Thematic template or guide? What was the focus in these groups, how was it facilitated and how were they conducted? Were the user representatives and informal caregivers in mixed groups with other stakeholders? If so, did you do any interventions to equalize the power relationship between the involved actors?	Figure 1

Our response: We thank the reviewer for this comment. We hope that the addition of Figure 1 already provides some clarity on the number of working groups and the themes they focused on. We did not use a structured interview guide, however, the topics discussed are mentioned shortly on page 12. We did mix the client representatives and informal caregiver representatives with other stakeholders. These people were professionals familiar with the field of research, therefore, we did not do any interventions to equalize power relationships. We agree with the reviewer that these are all important questions, however, because we are limited in words and are already exceeding the limit, we cannot elaborate extensively on these points in the manuscript.	
20. The last sentence in the methods section is “The results were discussed regularly within the research team” Who is the research team? Pleas provide some sentences about research reflexivity. Our response: We thank the reviewer for the comment and provided some more information about the composition of the research team: The research team consists of all authors involved. They work in the field of social gerontology, public health and primary care, and long-term care.	P 13
21. Please provide some sentences about research reflexivity. Our response: We agree with the reviewer that the information on reflexivity is limited. However, we did include some details on rigour; data triangulation, member checking, discussion within the research team to increase conformability. To add to the reflexivity, we added the use of a research log book which was used to keep track of research activities and intermediate results, and field notes of each activity: The research team consists of all authors involved. They work in the field of social gerontology, public health and primary care, and long-term care. A research logbook was kept by the researchers to keep track of all research activities and intermediate results, together with field notes from these activities.	P 13-14
Results	
22. A general objection is that the results are written in a way that does not make it visible that they are the empirical result of the study, which makes me question what you base your statements on. For example p. 12, line 21: “the program is delivered by an interdisciplinary team, generally consisting of an RN, an OT, and [...]” Is this results based on the study? Is it based on the literature review or the co-creation activities? Or is it just the authors assumption of general practice? This is not clear, and there are several examples of this throughout	Supplementary file 2 P 15

the results section. I think a rephrasing may help you with clarifying this. It is however important that the results you provide are actual results of this current study. More examples: "A care coordinator is appointed, in most cases, the OT [...]": Based on what?, "The reablement team may consult other care professionals" Based on what? (and how should that be done?), "(bi-) weekly team meetings are organized to [...]": Based on what?

23. *Was the content of this training program also a result of the co-creation process, or literature review? Again, this is a bit unclear.*
24. *You have based the initiatives on some standardized tools, such a COPM and OTAGO. What was the rationale behind this? Is it based on results from the co-creation process? Literature review? Or just a random choice?*
25. *Presentation of the I-MANAGE process: Again, I am having problems with reading this text as a presentation of study results. For example, the sentence at line 53; "People are referred to the program through community nurses [...]" What about writing: "a common perception in multi stakeholder discussion, was that referrals should be made by the community nurses", or "based on the literature review, it became clear that a referral from the community nurses is advantageously". Try to show the reader what evidence that support your claim.*
26. *Page 14, line 3-6: You claim that "The care coordinators initiates an exploratory conversation [...]", and then you refer to reference number 37. Are these results based on the literature review? Again, it is not clear what the evidence are.*

Our response:

We thank the reviewer for these comments and would like to provide clarity on how the results are presented. When writing up the manuscript, we initially tried to write the results as suggested by the reviewer. However, this did not seem to work because of the iterative nature of the development and the combination of different research methods. All results derive from a combination of methods and there is almost never one specific source for each statement/decision made. Moreover, presenting all the intermediate findings of the programme development steps in detail and linking them to the final programme components would make the result section very lengthy and would compromise readability. Our goal was to describe the content of the programme in detail, as well as the process of development. We therefore decided to present the results the way they are presented now; focussing on the content of programme. We did however mention the intermediate results in Supplementary file 3. This in combination with the addition of Supplementary file 2 hopefully provides more clarity on the choices made and the source they originate from.

We do understand why the reviewer mentioned this and we added a short introduction to the results section in order to provide some clarity on how the results were presented:

The following section presents the final model of the I-MANAGE programme in detail, starting with the developed logic model, and eventually the translation to practice, which contains a detailed description of all programme components. A

detailed description of the results from the first five programme development steps is provided in Supplementary file 3. We would also like to address some of the raised comments in more detail:  - The choice for using the COPM was mostly based on the scientific literature. However, we discussed the use of this instrument within the working group to assess if it would be feasible for them to include another instrument to their daily routine. - We integrated OTAGO based on experiences from previous research in the Netherlands with a reablement training programme, where OTAGO was experienced as a positive addition to usual home care. Additionally, it was also mentioned in literature a few times that this increased activity in older adults and could therefore be used to achieve older persons' goals. - The citation mentioned by the reviewer refers to the principle of Positive Health on which the exploratory conversation is based. 	
27. At p. 11, line 23-24 you write: "The intended results are reduced (I)ADL, [...]" I guess you mean that "the intended patient outcomes of reablement are [...]" This needs rephrasing so that readers do not get confused about the results of the study and the results/outcomes of the interventions. Our response: We thank the reviewer for the feedback. We adjusted the sentence to the following and also adapted it in Figure 2: The intended client outcomes of the programme are reducing [...]	P 9, p 15 and p, 16 Figure 2 p 15
28. P. 11: "Development of a logic model": This section is a bit confusing to read as it describes six stages of the method, three sessions conducted, six program components, and five phases of I-MANAGE. Can you find any ways to just make this easier for the reader to follow? Our response: We thank the reviewer for the comment. However, we are unsure what was meant with it. The section mentioned by the reviewer only describes the six programme components of the logic model, which is supported by Figure 2, and a reference to the 6 programme development steps to indicate where the elements of the logic model derive from. However, we hope that the addition of Figure 1 helps to distinct the steps, phases and components. Additionally, we explain in the manuscript that the steps refers to the methods. We also clarified that the 5 phases (described under 'Translation to practice: [...]') are a practical translation of the logic model:	P 15 and p 17

The logic model starts with the aim and sub-aims of the programme based on identified problems and needs (result from development steps 1 and 4), [...] – for all entries. The I-MANAGE care process consists of five consecutive phases: 1) initiation, 2) intake, 3) care plan, 4) care delivery, and 5) evaluation. Each phase is described in detail below and is presented in Figure 2. The five phases are a result of the practical translation of the logic model described earlier.	
29. At p. 13 you describe a practice-oriented training for the reablement staff. Who did/should provide this training? And how was the ‘detailed manual’ developed (and by whom?) and can the manual be made available for the reader in any way (if this is supposed to work as a ‘blueprint’ it should be made available). 30. At p. 14, line 39 you refer to an “available toolkit”. What toolkit is this? Can it be made available for the reader? Our response: We thank the reviewer for this comment. We would like to address the questions in details:  - Currently the training is given by researchers. However, in the future, when the programme is being implemented across other care organisations, it should be provided by trained trainers working for, e.g. a project leader or health care professional from the organisation. - The manual was developed by the researchers and revised by working group members to make sure it is suited for practice. As described in the manuscript, the manual consists of background information and a description of the programme, including the goal, key components, and I-MANAGE’s care process. It is a more detailed and practice oriented version of the programme as written in the manuscript. We added this information to the manuscript: [...] The manual was developed by the researchers and revised by working group members to make sure it is suitable for practice. [...]  - As mentioned before, the manual including the toolkit has to be tailored to organisations depending on their resources, target group, etc. Interested readers can contact the author for more information. 	P 17
Discussion	
31. You state that the model leaves room for contextual adjustments. I miss a description of how this adjustment was done in this current study, concerning the Dutch context. Our response: We thank the reviewer for the comment. We would like to point out that during the development process described in the manuscript (elements of) the programme were not yet executed. At the moment the programme is being pilot tested in two	P 22 and p 23

care organisations in the Netherlands. We do provide some examples of how the programme can be tailored to fit the context and organisation in the discussion section (page 22). Additionally, at the end of the discussion, we mention that the programme is currently being evaluated (page 23).	
32. You also state that an example of adjustment can be to utilize other professions or care staff. Does it matter what professions that provide the model? What was the rationale behind utilizing OTs and RN in this study? Our response: We thank the reviewer for this comment. This choice to include both, OTs and RN, is based on the literature and the fit with Dutch community-care system. First literature states that including allied health in the reablement team is beneficial. We also mention this in our discussion on page 20. Making the team interdisciplinary, for example by including RN, also has its benefits as mentioned in the discussion. We choose to include RN because they have a key role in the Dutch community care setting: district nursing teams consist of approximately 10–15 staff members: bachelor-educated registered nurses, vocationally trained registered nurses, (certified) nurse assistants and nurse aides. The district nurse, who usually has a bachelor's degree, leads the nursing team, coordinates and supervises care delivery, and is responsible for, among other things, conducting formal needs assessments and developing care plans. Because of this, the working group members indicated that the RN has to be a key person within the reablement team. However, the team composition can vary, as stated in the manuscript, depending on available resources and whether that specific discipline is present in the organisation or not.	P 20
33. At p. 18, line 39-43 you describe results from a scoping review, and refer to the review, but also an additional study, which is not the review. Why is this? Our response: We thank the reviewer for this comment. The reference is mentioned as an additional source to support the information written. However, we do understand the confusion and therefore deleted the reference.	P 22-23
Response to concerns raised by reviewer 2	
34. It was a pleasure to read your manuscript which I hope will soon be published after some revisions. It is an important contribution to the field of reablement based on an ambitious co-design approach. Our response: We thank the reviewer for the compliments.	N/A
35. A general comment that you might consider in the first sentences is that the language can be interpreted as slightly ageist. While disability becomes more frequent as people age, a large proportion in the older population in fact remain independent and can live an active life. For people beginning to experience	P 6

disability, it is often minor disability from the beginning. As the text is written now, the reader can get the impression that most older people will suffer from severe disability. See for example Gore or Zingmark that disability rather can be emerging over time. https://academic.oup.com/ageing/article/47/6/764/5079486 https://bmcgeriatr.biomedcentral.com/articles/10.1186/s12877-021-02283-x Our response: We thank the reviewer for this comment and for pointing out the language use. We agree that this might be considered ageist, therefore we adapted the first sentences of the 'Background' to the following: The amount of older adults experiencing disabilities will increase over time, and while a large proportion of the older population remains independent, others will experience an increased need for support [5]. Moreover, 50% of people aged 85 years or older require care and/or support with daily activities [3]. As a result, it is expected that their demands for long-term care services will increase. When older people live in an environment that is not safe and does not meet their needs, the challenges they might face regarding independent living will become even bigger [4].	
36. In the end of the first paragraph: "Not tackling these challenges properly could lead to increased healthcare utilization and costs [13], and unnecessary (re)hospitalisations or permanent nursing home placement..." I think it also could be increased use of social care services (it would be home care in a Swedish context). Please adjust if this applies to the context of your country. Our response: We agree with the reviewer and adjusted the sentence to the following: Not tackling these challenges properly could lead to increased utilisation of health and social care and related costs [14]	P 6
37. In general, I agree with your statement that "reablement programmes, are often insufficiently described in the scientific literature. But not always, see for example these references: https://www.tandfonline.com/doi/full/10.1080/11038128.2022.2089229 https://bmcgeriatr.biomedcentral.com/articles/10.1186/s12877-022-03185-2 https://onlinelibrary.wiley.com/doi/abs/10.1111/hsc.12934 Adding your paper, once published, the literature is becoming increasingly	P 7

better to describe the actual content of reablement and how it is delivered. Our response: We thank the reviewer for pointing out these references. We included some of them in our background section as follows: There are only limited articles available that describe either the development of the programme or its content in detail [32-34], and often these descriptions are included as part of a feasibility or pilot study [30].	
38. The last paragraph is quite long which lead to that the study objective is not 100% clear. Could some information go into the methods and even discussion (“Due to its extensive description, this manuscript may serve as a blueprint for implementation in other local settings?”). Our response: We agree with the reviewer that some of the information in the last paragraph was unnecessary and was already mentioned extensively in the methods. We therefore revised the last paragraph to the following: Therefore, this paper describes the development, using a co-creation process, and content of I-MANAGE, a reablement programme for community-dwelling older adults to improve older persons’ self-management and participation in daily life, while also increasing quality of life and decreasing informal caregivers’ burden. By describing the development and content of the programme in detail, we increase the replicability and prevent other researchers from re-inventing the wheel. The programme is specifically suitable for the Dutch community care setting, however, due to its extensive description, this manuscript may serve as a model for implementation in other countries as well.	P 8
39. Patient and public involvement: Who were “All end-users”? Were it those who were to implement the program or the target population who needs were to be addressed. Or both? Our response: We thank the reviewer for this comment and added a short description of the end-users under ‘patient and public involvement’: All end-users (care professionals who would be implementing and delivering the programme, as well as the target population) were represented during and involved	P 9-10

in the development process of the programme by being a member of the working groups and participated in the individual interviews. They were not involved in the development of the study design of dissemination of the findings.	
40. Setting and participants: You write: "The programme was developed for community-dwelling older people who wish to remain living at home independently". First, doesn't almost all older people want to do that? In that case, did you have more specific criteria? 41. I wonder why I-MANAGE is intended for community-dwelling older people? Isn't reablement a service intended for clients irrespective of age (see definition based on Delphi study) Our response: We thank the reviewer for these comments. We agree that most older people would like to remain at home for as long as possible. Thus this information has no added value here. With regard to the target group, we agree that the philosophy of reablement is inclusive for all clients, but we think that an in-patient setting requires a different approach, as people have long-term care needs and care is provided 24/7.	N/A
42. In this paragraph, it is not clear how participants were recruited and informed about the study (check how this match with your section about ethics). Our response: We thank the reviewer for this comment. We added a few sentences to provide clarity on where participants were recruited from and how they were informed, this is also in line with comment 15 raised by reviewer 1: Participants were recruited from the professional network of the researchers. They were informed about the study and asked to participate via email. When participants agreed to participate, verbal or written informed consent was provided before the start of each interview or working group session.	P 10
43. There were working groups. How many? Who were included in which working group? How often did they meet and over how long time period? Or was it only one working group? Our response: We thank the reviewer for the comment. We would like to refer to comment 15 raised by reviewer 1. Additional information about the different stakeholders per research activity is provided in Supplementary file 2. The table was included as a supplementary file due to the journal's guidelines for tables. Additionally, we would like to refer to comment 19 raised by reviewer 1, where also more information about the process was requested. As a response, we created	Figure 1 p 13

Figure 1 which describes the development process in more detail.	
44. As I read, I understand that the program was developed in one local context. Could you provide some additional information about this city or municipality, population size, was it similar to other contexts in your country? Since you later argue that a programme like I-MANAGE can be tailored to local contexts, here could be a place to provide a good basis for this line of reasoning. Our response: We thank the reviewer for the comment. We would like to refer to comment 12 raised by reviewer 1. We provided additional information about the Dutch home care context as a response to this comment: I-MANAGE is based on international evidence and tailored to the Dutch home care context. Home care in the Netherlands includes personal care (i.e. assistance with activities of daily living (ADL)), nursing care (i.e. medical assistance), and domestic support (i.e. assistance with instrumental activities of daily living (IADL)) [42]. Usually home care is funded by two statutory forms of insurance cover care: the Health Insurance Act (ZVW), and the Social Support Act (WMO). Clients often use a combination of ZVW (e.g. general practitioner care, therapists, hospital care, or medication), and WMO (e.g. domestic support, home adaptations) [42, 43].	P 10
45. I agree that ethics were not required for this study. However, you do not have any information about participant characteristics which is a limitation. Did you do any efforts to ensure variation among those recruited e.g., in terms of age and gender, experience of reablement... Please write something about this and if not anything was done, name it as a limitation. Our response: We thank the reviewer for this comment. We did not do anything to ensure variation among those recruited, as mentioned previously, all participants were recruited from the professional network of the researchers and happened more conveniently. We therefore included this as a limitation of our study: Additionally, the care professionals were recruited from the professional network of the researchers and were recruited in a convenient way. We did not ensure variation among these participants in terms of, for example, gender, age, or years of experience.	P 24
46. Development of a logic model (1-6):  a. Who led the working groups? b. There were rather large groups (9 participants + researcher). Did you do anything to ensure that all participants had the chance to be involved in the co-design process? c. Was the group size a limitation? 	P 12

Our response: We thank the reviewer for these comments. The working groups were led by two researchers of the research team. We added this to our methods: [...] The working group sessions were led by the first and last author (IM and SM). We understand that group size may be a limitation and that there could be a possibility that not all participants had the chance to be involved in the co-design process. However, we did not experience this during our development process.	
47. You mention the steps, but it is confusing that you do not mention them in order 1, 2, 3,... Our response: We thank the reviewer for this comment. We would like to refer to comment 19 raised by reviewer 1, where we explain that the model we based our development on (from Bleijenberg et al) presents a non-linear and iterative development process. This, combined with the variety of methods we used, made it difficult to present it chronologically. We hope that the addition of Figure 1 provides more clarity on the process.	Figure 1 p 13
48. In addition, it is also unclear when you performed this: “Lastly, observations were performed to examine current practice and context of home care services (step 5) and to identify problems (step 1). Six observations of half a day each were performed with a DSW, an RN, and a nursing assistant, and three days with allied health. Our response: We thank the reviewer for this comment. As a response we would like to refer to the addition of Figure 1 which describes the development process more in detail.	Figure 1 p 13
49. The first step is to identify the problem. I understand that “the problem” is what the program will address but could you explain that so it is VERY clear. 50. You do a lot to get an accurate picture of the problem, right? However, the logic presentation of what you have done and when you did it isn’t completely clear. I suggest something in line with: To ensure a valid identification of the problem we conducted observations, individual interviews discussions in the working groups....., All subsequent steps was based on the identified problem.... Our response: We thank the reviewer for these comments. We would like to refer to comment 50, where the presentation of the steps was also discussed. As mentioned before, the development of the programme was a non-linear and iterative process, making it impossible to describe the data collection methods in order of the steps (1 – 6). As	Supplementary file 2

a research team, we chose to describe the data collection per methodology used (literature, working groups, observations, and interviews), because this would result in less repetition. To explain which methodology was used for each step, we added Supplementary file 2. The table was included as a supplementary file due to the journal's guidelines for tables.	
51. Was it the same people that were included in the 11 interviews? If yes, please clarify. If no, please describe how they were recruited. 52. Step 7: You write: "For example, on informal caregiver support, we invited an informal care consultant, an informal caregiver representative, and a psychologist; resulting in three to four members per working group." It is difficult to understand how many working groups there were. Can you please specify that? And again, who led the working groups, the same person or different persons? Our response: We thank the reviewer for this comment. We would like to refer to comment 15 raised by reviewer 1. Additional information about the different stakeholders per research activity is provided in Supplementary file 2. We would also like to refer to comment 45 where we added information about the recruitment process.	Supplementary file 2 and p 10
53. Data analysis: Did you also included data from ongoing communication: "Between sessions, working group members were consulted for additional input" Our response: We thank the reviewer for this comment. We would like to refer to comment 18 raised by reviewer 1, where more information was asked about what was done in between sessions. We added the following to the manuscript: Between sessions, working group members were consulted for additional input and clarification if needed. The researchers processed the results from each session and the additional information in order to be used as a starting point for the next session.	P 11-12
54. Clearly stating what the problem is. I think the text would benefit much if you should start with presenting what the identified problems were. Then, it will be much easier to understand why you decided to include the six programme components. It is clear in supplementary file 2 but I think the reader should access this information much more easily since it is so central for your results. Our response:	P 15

We thank the reviewer for this comment and suggestion. We would like to refer to comment 25 raised by reviewer 1, where we explain why the results are presented this way. Our goal with this paper was to focus on the development and content of the programme. Therefore we presented the results as focusing on the content of the programme. As mentioned by the reviewer, we included the intermediate results of each step in Supplementary file 3. We also refer back to the problems in the logic model under Aim (Figure 2). However, we understand that these intermediate results could be of added value to the reader, therefore, we refer to Supplementary file 3 earlier on in the result section: [...] A detailed description of the results from the first five programme development steps is provided in Supplementary file 3. This comment refers to Supplementary file 3 due to the addition of another supplementary file.	
55. Further, in supplementary file 2, you say that “There appeared a gap between the person’s abilities, needs, and wishes, and the environment they reside in. When older people experience a deterioration in health status, their environment is often not adapted, i.e. the necessary home modifications and assistive devices are not always in place” You refer to Wahl et al from 2009. First, I think the reference is a bit old, is there something more up-to date? Second, I wonder if Wahl et al., refer to a similar population as for which your programme was developed? Please consider carefully is this reference is appropriate. Our response: We thank the reviewer for this comment. We agree that Wahl et al 2009 is not appropriate here. We noticed that in the supplementary files something went wrong with the references (as well as the page numbers as mentioned in comment 71). We adapted the references in both supplementary files. They should be more appropriate now. We would like to point out that the page numbers and reference numbers in the supplementary files refer to the clean version (not with track changes). Due to the use of referencing software, the reference list did not update accordingly and therefore, there could be some discrepancies. This comment refers to Supplementary file 3 due to the addition of another supplementary file.	Supplementary file 3
56. I miss a clear structure that follows the logic of the model. I think that the logic can provide an easy structure that can help the reader understand step by step what you found. Please consider to use the logic and the order 1-7 not only in the supplementary file. Step 1 identifying the problem Step 2 previously identified evidence Step 3 theoretical foundation for the programme Step 4 determination of	N/A

needs Step 5 examination of the current practice and context Step 6 modelling the process and outcomes Step 7 translate components into practice Our response: We thank the reviewer for this comment and suggestion. We would like to refer to comment 25 raised by reviewer 1, where we explain why the results are presented this way.	
57. There is a discrepancy which I think could be made clearer. You write: “I-MANAGE is intended for community-dwelling older people” and later “I-MANAGE aims to improve interdisciplinary collaboration”. Thus, is the programme designed to improved delivery of reablement? That is, is it designed to help professionals to initiate and deliver the service, which in turn benefits the clients? Our response: We thank the reviewer for this comment and agree that the wording is not quite right. The programme is indeed intended for community-dwelling older people. We changed it to the following: I-MANAGE facilitates interdisciplinary collaboration [...]	P 16
58. Why is the duration 8 weeks? Was this a result of the co-design process? Our response: We thank the reviewer for this comment. The 8 weeks is partly a result of the co-design process. We know from the literature research that we performed during this study that the average duration of reablement interventions lies between 6 and 12 weeks. When discussing these results with the members of the working groups, they agreed that 8 weeks would be sufficient. However, as mentioned previously, the programme was at that moment not yet tested in practice. The programme is now being pilot tested and we have to evaluate if this duration is sufficient.	N/A
59. OTAGO is included and usually this is implemented to reduce falls. However, it is not clear from the identification of problems, determining needs or modelling outcomes that fall-related injuries is something that needs to be more specifically addressed in reablement. Is that information missing? Or can you elaborate the text why OTAGO needed to be included? In addition, it is not clear if and how OTAGO is implemented when delivering the programme (phase 4). Also in figure 1, any text on OTAGO is missing. Our response: We thank the reviewer for the comment. We would like to refer to comment 27	P 17

raised by reviewer 1. We integrated OTAGO based on experiences from previous research in the Netherlands with a reablement training programme, where OTAGO was experienced as a positive addition to providing daily care. Additionally, it was also mentioned in literature a few times that this increased activity in older adults and could therefore be used to achieve older persons' goals. What we did in this programme (but also the previous research mentioned earlier) was we created an exercise booklet based on the OTAGO programme as mentioned on p 17. The booklet is part of the toolkit for achieving individual client goals. That is also the reason it is not mentioned in Figure 2 (description of the programme), because it is not used in every client, it depends on the goals of the client.	
60. "The training consists of multiple sessions". Can you be more specific and write exactly how many sessions were included and over how long time training was implemented. After how long time was the booster session? Our response: We thank the reviewer for the comment. As described in the manuscript, the training consists out of a minimum of 4 sessions (kick-off meeting, booster session, and 2 specific sessions on the use of the COPM and OTAGO). However, the amount of sessions and timing of these session also depends on the resources of the organisation implementing the programme. We would also like to refer to comment 26 raised by reviewer 1, where more information on the training was requested.	P 17
61. Phase 5: "The evaluation of the care process is structurally embedded in interdisciplinary team meetings". How often are these meetings? Our response: We thank the reviewer for this comment. We would like to refer to the section on 'Interdisciplinary collaboration' on p 16, where we describe that (bi-)weekly meetings are organised within the team to discuss the intake of new clients and informal caregivers, the progress made by clients, and final evaluations of clients' personal goals.	P 16
62. When reading the discussion, I get the impression it could be elaborated further. The authors state "The described programme serves as a blueprint and leaves room to tailor the intervention to a specific context". Although I agree, I do not think the results or methods emphasized adaptation to the local context sufficiently to raise highlight this early in the discussion. I rather miss a distinct discussion of the findings presented in this study compared to previous research. 63. Also here, the logic of the model could provide a structure to reflect over the results and the methods used For each of the following, to what extent was your findings similar or different to previous research? Step 1 identifying the problem Step 2 previously identified evidence	P 20-24

Step 3 theoretical foundation for the programme Step 4 determination of needs Step 5 examination of the current practice and context Step 6 modelling the process and outcomes Step 7 translate components into practice Another or an additional perspective could be to discuss the five phases in the I-MANAGE care process; to what extent was your findings similar or different to previous research? Our response: We thank the reviewer for these comments and agree that the adaptation to a local context is not appropriate in the beginning of our discussion. Therefore we moved this section towards the end, of the discussion where we also reflect on how to implement the programme. Additionally, we agree that discussing our findings/content of the programme compared to other programmes would benefit the discussion section. We already reflected on the key elements of our programme in relation to other programmes and previous research on reablement (assessment and goal-setting, interdisciplinary collaboration, training, team composition, etc.). In the revised manuscript we added some information to these paragraphs and changed the order so that it follows the same order as the results. We decided to follow the order of the phases as following the structure of the development model would not be feasible as mentioned previously.	
64. Further, as described in the methods, steps 1-6 were about development of the programme whereas step 7 was about translating and choosing relevant actions for optimizing delivery of the programme. What was the pros and cons of this approach? Were there similarities or differences when comparing to other programmes that were based on co-creation. See for example https://www.tandfonline.com/doi/full/10.1080/11038128.2022.2089229 in which there are some similarities but also clear differences in (a) the scope of stakeholders involved in the co-design process and (b) the scope of the programme. However, there was similarities in the phases described. Our response: We thank the reviewer for this comment and the suggested literature. We are aware that there are multiple approaches to develop a complex intervention, all with their own pros and cons. We made the decision to use the approach described by Bleijenberg et al from 2018 because this fitted best within our entire research project which is based on the MRC framework. As mentioned by the reviewer, the different approaches have a lot of similarities and no approach is set in stone and leaves room for the researchers own interpretation. It also has to fit the setting and timing of the development process. We agree that this information is not mentioned in the article, therefore we reflected on the use of this approach in the	P 23-24

strengths/limitations paragraph of our discussion: We used the development approach described by Bleijenberg et al. [41], which combined a range of published approaches to intervention development to enrich the MRC framework. This approach was chosen because using the MRC framework would help us in further evaluating and adapting the programme. However, we are aware that multiple approaches to intervention development exist as described by O’Cathain et al. [66]. These different approaches share many similarities (e.g. stepwise approach or involvement of stakeholders), but there are also significant differences (e.g. the focus on implementation or theory). It is important to acknowledge these differences and always chose an approach best suited for the purpose of the research. Additionally, most of these approaches are not set in stone and leave room for the researchers own interpretation. It also has to fit the setting and timing of the development process.	
65. In addition, different tenses (past, present) are used. Our response: We thank the reviewer for this comment and checked the use of tenses critically. We noticed that both past and present tenses are use. However, because we describe a programme that still exists and is currently being evaluated, all parts of the manuscript referring to the programme or its content, are written in present tense. The parts referring to the development process or research that has been done previously, is written in past tense.	N/A
66. There are several references that are a bit old (10 years or more) and therefore you may consider to update some references, see suggestions in previous comments. Our response: We thank the reviewer for this comment. We critically reviewed the use of our references. We agree that some are a bit old, therefore we adjusted some of them, and made sure that all publications that were over 10 years old are accompanied by a more recent reference. Two references referring to MRC and COPM are key publications and cannot be updated.	Throughout manuscript
67. Figures are not numbered. Our response: We noticed that the numbering of the figures is not visible after uploaded the documents separately. However, since the figure must be uploaded without caption and legend, it was not possible to adapt this. The legend of each figure can be found at the end of the manuscript (as required by the guidelines). The figures are uploaded in order, hopefully making it easier to follow.	N/A
68. Do the page numbers referred in the table match the actual page number in the document? I find it hard to follow.	Supplementary file 1

Our response:

We thank the reviewer for noticing. We adapted the page numbers in the table to match the manuscript (the page numbers and references are from the clean version, not tracked changed).

VERSION 2 – REVIEW

REVIEWER	Marianne Eliassen UiT The Arctic University of Norway Faculty of Health Sciences
REVIEW RETURNED	03-Apr-2023

GENERAL COMMENTS	Thank you for doing a thorough revision of this manuscript. The work that you are reporting on is both complex and multi-faceted, and I understand that it is a challenging exercise. However, the revision has made things much clearer for me! I still have some objections that I believe should be addressed to make the manuscript ready for publication. 1. The introduction section is now both clear and thorough. Including research from an international context – and describing how the Dutch context may differ, clarify the need for your study.2. Methods: We (both reviewers) asked for a clearer description of how the Logic model-development was embedded in the process. I think this is clearer now. However, there have been some replications in the text due to the revision. You write about the Logic-model both in the “design-section”, and then again later in the section called “Development of a logic model (steps 1 to 6)”. Try to arrange the text so that you avoid repetition. My suggestion is to wait with the presentation of the Logic model till after you have described the setting, participants and the co-production (otherwise, one may believe that the logic model was conducted by the researchers in advance of the co-creation).3. Methods: The description of the data analysis is shallow. You state that the working group sessions were the main source of information, but all data sources were triangulated. What do you mean by that. Was it because the group sessions represented the largest amount of data, or was it considered to be more important? (How did you do the triangulation?) Did you base the thematic analysis on any methodological theory? Do you have any references that support your choice of approach.4. My main objection is still how the results are presented. This was my main objection in the first round of revision, and I still believe that this must be revised in order to make the article ready for scientific publication. The way that the results are currently presented, makes it look like the I-MANAGE program already is implemented, and that you describe existing practices in I-MANAGE. As I can understand, this is not the case, and this way of presenting the results is in that matter a flaw. Examples: “I-MANAGE facilitates interdisciplinary collaboration by facilitation intensive collaboration between ...” and; “[...] when discharged from another care facility [...] a smooth handover of client information is initiated so no necessary details of the client’s care process are lost” This may lead the reader to believe that you have scientific evidence for increased collaboration, or uninterrupted information handover in the I-MANAGE program. You have not assessed this and cannot conclude about it either. (These excerpts are only examples, and all results are presented in
---

	this way, and should be revised). To my understanding, your results provide knowledge on a normatively level: how things should be (the model/procedures). Maybe using wordings as “should”/ “aim to”/ “intend to” can help you with re-wording your results in line with what you actually can say something about. This comment is in line with my comments from the first round of review, which was not accommodated, and I still believe that this is an important issue in this paper.
--	---

REVIEWER	Magnus Zingmark Municipality of Östersund, Health and Social Care Administration
REVIEW RETURNED	11-Apr-2023

GENERAL COMMENTS	Overall impression I think the revised text has improved the quality of the paper. As I have said before, this paper is an important contribution to the field of reablement based on the ambitious co-design approach. However, I suggest some revisions which I think would improve the paper further. In addition, I wonder of the revised version has been checked with regards to the language. In a few places I wonder if the language is OK. I look forward to read a revised version of the manuscript. Methods, Design I think the authors should refer to Figure 1, the first time it is mentioned. Thus, under Design. In some way with the text first presenting the logic model (under Design). Keeping all descriptions of these steps on one place will make the text easier to read. Methods, Data collection Could the Data collection be merged with the Data analysis section? Results The following sentence is long. If divided in two sentences it could become clearer. “The intended results client outcomes of the programme are reducing (I)ADL disability, improving self-management skills, increasing the quality of life of both the client and informal caregiver, and reducing healthcare utilisation and expenditures (proximal outcomes), which are common outcome measures in reablement programmes abroad, to remain living at home independently and avoid unnecessary transitions to institutional care (distal outcomes).” The average duration of the programme is 8 weeks. Why is it time limited? Is it part of the results, e.g., that stakeholders considered 8 weeks to be sufficient or some other reason? In addition, when you say it was the average time, have you tested it and calculated the average? If so, please add information on that. Or, if 8 weeks rather was an approximation. In the following sentence, HOW does the programme facilitate intensive collaboration? This is always challenging so what you do is important to describe. Maybe you could refer to the structured approach including the phases described a few paragraphs below? “I-MANAGE aims to improve facilitates interdisciplinary collaboration by facilitating intensive collaboration” Phase1: “The care coordinator plans a first visit to present the programme to the client and their informal caregiver”. Is there always an informal caregiver?
---

	Phase2: “the OT performs an environmental assessment”. Is this limited to the environment inside the home or does it also include entrance and outdoor environment? Phase 4: Can you say anything about the intensity, e.g., the number of visits per day or week that can be included? Such information is very relevant for understanding the resources required if the programme should be used in another context as you suggest. Phase 5: “At the end of the programme, a formal evaluation of the client’s goals takes place using COPM”. Does this also include the scoring, i.e., using the same procedures as on the first occasion? Discussion Overall, the authors have developed the programme in a very structured approach. However, the overall tense in the text may be interpreted as if I-MANAGE WILL have positive effects, but doesn’t that have to be explored? Even if the program includes several components that seem promising and that can be considered evidence based, I suggest that the authors highlight the need for further trials to explore feasibility and evaluation of effectiveness. Below are some examples to clarify my reasoning. Example on p21 (version with track changes): “Previous research has indicated that using standardised assessment or goal-setting tools could increase the effectiveness of reablement interventions [32]. Additionally, it increased client involvement and helps professionals to identify meaningful activities with the client [56].” While I-MANAGE include these components, it still remains to be explored whether it results in positive effects. In the discussion about the COPM you only refer to positive findings but reference 35 indicate that COPM might be challenging to use “by the book” (e.g. including scoring) for 100% of all clients. Please consider if you like to frame it as you do in the results “Clients MUST score both their performance and their satisfaction with performing these activities”. What if some clients can’t engage in the scoring procedure? In the following sentence it seems as you know that I-MANAGE will be successful. Before preliminary testing and rigorous evaluation can you actually claim the following? “First, supporting informal caregivers also contributes to the success of I-MANAGE.” I do agree that it is an important component of an intervention program! In addition, success is a strong word and if it is to be used a suggest the authors specify what they mean with success. In the sentence “For example, the programme leaves room to the organisation to choose which target group would benefit the most.” An organisation can choose how to implement a programme like I-MANAGE. However, wouldn’t it be relevant to explore the question whether I- MANAGE result in positive effects and whether these effects differ between groups? That would be even more helpful for organisations when considering to implement reablement.
--	---

VERSION 2 – AUTHOR RESPONSE

Response to concerns raised by reviewer 1	
Introduction	
69. The introduction section is now both clear and thorough. Including research from an international context – and describing how the Dutch context may differ, clarify the need for your study. Our response: We thank the reviewer for the compliments.	N/A
Methods	
70. We (both reviewers) asked for a clearer description of how the Logic model-development was embedded in the process. I think this is clearer now. However, there have been some replications in the text due to the revision. You write about the Logic-model both in the “design-section”, and then again later in the section called “Development of a logic model (steps 1 to 6)”. Try to arrange the text so that you avoid repetition. My suggestion is to wait with the presentation of the Logic model till after you have described the setting, participants and the co-production (otherwise, one may believe that the logic model was conducted by the researchers in advance of the co-creation). Our response: We thank the reviewer for this comment. We moved the information about the logic model to ‘Data collection – Development of a logic model (steps 1 – 6)’.	P 9 - 10
71. The description of the data analysis is shallow. You state that the working group sessions were the main source of information, but all data sources were triangulated. What do you mean by that. Was it because the group sessions represented the largest amount of data, or was it considered to be more important? (How did you do the triangulation?) Did you base the thematic analysis on any methodological theory? Do you have any references that support your choice of approach. Our response: We thank the reviewer for this comment. We aimed to develop a model that was in line with the needs and wishes of our target groups, therefore we consider the working group sessions to be our main source of information. We mentioned in the ‘Data analysis’ part that we use the individual interviews, observations, and literature research to complement and check the obtained information from the working group sessions. However, this could be mentioned sooner to provide more clarity, hence, we adapted this in the text. Regarding the thematic analysis, we did base our methodology on the steps designed by Braun & Clarke. We added this to our methods. We used data triangulation to verify the results. Our main source of information was the working group sessions, as they provided the richest data on the perspectives	P 12

of the different target groups. Individual interviews, observations, and literature research were used to complement and check the information obtained throughout the working group sessions. Working group sessions and individual interviews were recorded and transcribed non-verbatim. A thematic analysis was conducted based on the steps described by Braun and Clarke [44].	
Results	
72. My main objection is still how the results are presented. This was my main objection in the first round of revision, and I still believe that this must be revised in order to make the article ready for scientific publication. The way that the results are currently presented, makes it look like the I-MANAGE program already is implemented, and that you describe existing practices in I-MANAGE. As I can understand, this is not the case, and this way of presenting the results is in that matter a flaw. Examples: "I-MANAGE facilitates interdisciplinary collaboration by facilitation intensive collaboration between ...]" and; "[...] when discharged from another care facility [...] a smooth handover of client information is initiated so no necessary details of the client's care process are lost" This may lead the reader to believe that you have scientific evidence for increased collaboration, or uninterrupted information handover in the I-MANAGE program. You have not assessed this and cannot conclude about it either. (These excerpts are only examples, and all results are presented in this way, and should be revised). To my understanding, your results provide knowledge on a normatively level: how things should be (the model/procedures). Maybe using wordings as "should"/ "aim to"/ "intend to" can help you with re-wording your results in line with what you actually can say something about. This comment is in line with my comments from the first round of review, which was not accommodated, and I still believe that this is an important issue in this paper. Our response: We thank the reviewer for this comment and the suggestion on how to revise the result section. We adapted the results accordingly.	Results P 14 - 19
Response to concerns raised by reviewer 2	
73. I wonder of the revised version has been checked with regards to the language. In a few places I wonder if the language is OK. Our response: We thank the reviewer for this comment. The paper was checked by a native speaker and necessary changes were made.	Throughout manuscript.
Methods	
74. I think the authors should refer to Figure 1, the first time it is mentioned. Thus, under Design. In some way with the text first presenting the logic model (under Design). Keeping all descriptions of these steps on one place will make the text easier to read.	P 10

Our response: We thank the reviewer for this comment. We would like to refer to comment 2 raised by reviewer 1. We moved some of the information to the 'data collection' section of the methods. The first reference to Figure 1 is mentioned here. We did move the placement of Figure 1 to the same section	
75. Could the Data collection be merged with the Data analysis section? Our response: We thank the reviewer for this comment. We prefer to separate the 'Data collection' from the 'Data analysis' section because we believe merging the two would compromise readability. Moreover, the data analysis section is overarching for both phases of the data collection ('development of the logic model' and 'translation to practice') and merging the two would results in repetition.	N/A
Results	
76. The following sentence is long. If divided in two sentences it could become clearer. "The intended results client outcomes of the programme are reducing (I)ADL disability, improving self-management skills, increasing the quality of life of both the client and informal caregiver, and reducing healthcare utilisation and expenditures (proximal outcomes), which are common outcome measures in reablement programmes abroad, to remain living at home independently and avoid unnecessary transitions to institutional care (distal outcomes)." Our response: We thank the reviewer for this comment. We adapted the text to the following: The intended client outcomes of the programme are reducing (I)ADL disability, improving self-management skills, increasing QoL of both the client and informal caregiver, and reducing healthcare utilisation and expenditures (proximal outcomes), which are common outcome measures in reablement programmes abroad. Eventually, improving these proximal outcomes would help the older person to remain living at home independently and avoid unnecessary transitions to institutional care (distal outcomes).	P 14
77. The average duration of the programme is 8 weeks. Why is it time limited? Is it part of the results, e.g., that stakeholders considered 8 weeks to be sufficient or some other reason? In addition, when you say it was the average time, have you tested it and calculated the average? If so, please add information on that. Or, if 8 weeks rather was an approximation. Our response: We thank the reviewer for this comment. The 8 weeks is partly a result of the co-design process. We know from the literature research that we performed during this study that the average duration of reablement interventions lies between 6 and 12 weeks. Additionally, in general reablement programmes are time-limited. When	P 15

discussing these results with the members of the working groups, they agreed that 8 weeks would be sufficient. We did not test and calculate the average since, as mentioned previously, the programme was at that moment not yet tested in practice. The programme is now being pilot tested and we have to evaluate if this duration is sufficient. We understand that the wording might be confusing and therefore, we would also like to refer to comment 4 raised by reviewer 1. Here we adjusted the wording of the results section, including the part on duration: The average duration of the programme should be 8 weeks according to the members of the working groups.	
78. In the following sentence, HOW does the programme facilitate intensive collaboration? This is always challenging so what you do is important to describe. Maybe you could refer to the structured approach including the phases described a few paragraphs below? "I-MANAGE aims to improve facilitates interdisciplinary collaboration by facilitating intensive collaboration" Our response: We thank the reviewer for this comment. However, in our opinion the section on 'interdisciplinary collaboration' does describe the activities that were integrated in the programme to facilitate the collaboration. For example, delivering the programme by an interdisciplinary team, appointing a care coordinator, informing and coaching 'external' care providers, organizing (bi-)weekly team meetings, having access to a shared electronic care file, and initiating a smooth handover of information. We do not see how we can explain this more and hope this is sufficient.	P 15-16
79. Phase 1: "The care coordinator plans a first visit to present the programme to the client and their informal caregiver". Is there always an informal caregiver? Our response: We thank the reviewer for this comment and would like to clarify that there is not always an informal caregiver. We adjusted this in the text: The care coordinator plans a first visit to present the programme to the client and, if applicable, their informal caregiver	P 17
80. Phase 2: "the OT performs an environmental assessment". Is this limited to the environment inside the home or does it also include entrance and outdoor environment? Our response:	P 17

We thank the reviewer for this question. The environmental assessment is not limited to the inside environment and does include entrance and outdoor environment. We added this to the manuscript for clarification: Within the first week after the initiation phase, the OT should performs an environmental assessment, identifying necessary home modifications and assistive devices to ensure a safe environment. The environmental assessment is not limited to the inside environment but does also include the entrance and outside environment.	
81. Phase 4: Can you say anything about the intensity, e.g., the number of visits per day or week that can be included? Such information is very relevant for understanding the resources required if the programme should be used in another context as you suggest. Our response: We thank the reviewer for this comment and agree that this is very relevant information. Unfortunately, as mentioned in the description of phase 3, we cannot say anything about the intensity. This depends on the care needs of the client and their pre-set goals, which may require a higher intensity in the beginning but less at the end when (sub-)goals are (partly) reached.	N/A
82. Phase 5: “At the end of the programme, a formal evaluation of the client’s goals takes place using COPM”. Does this also include the scoring, i.e., using the same procedures as on the first occasion? Our response: We thank the reviewer for this comment. The formal evaluation at the end also includes the scoring. This is according to the protocol of the COPM, which includes two time points for scoring so the differences in scores can be calculated. We added more clarification to the text: At the end of the programme, a formal evaluation of the client’s goals should take place using COPM [46], including scoring the performance and satisfaction within activities.	P 19
Discussion	
83. Overall, the authors have developed the programme in a very structured approach. However, the overall tense in the text may be interpreted as if I-MANAGE WILL have positive effects, but doesn’t that have to be explored? Even if the program includes several components that seem promising and that can be considered evidence based, I suggest that the authors highlight the need for further trials to explore feasibility and evaluation of effectiveness. Below are some examples to clarify my reasoning. 84. Example on p21 (version with track changes): “Previous research has indicated that using standardised assessment or goal-setting tools could increase the effectiveness of reablement interventions [32]. Additionally, it increased client	P 22

involvement and helps professionals to identify meaningful activities with the client [56].” While I-MANAGE include these components, it still remains to be explored whether it results in positive effects. 85. In the sentence “For example, the programme leaves room to the organisation to choose which target group would benefit the most.” An organisation can choose how to implement a programme like I-MANAGE. However, wouldn’t it be relevant to explore the question whether I- MANAGE result in positive effects and whether these effects differ between groups? That would be even more helpful for organisations when considering to implement reablement. Our response: We thank the reviewer for this comment and agree that more emphasis on the need for further research should be added. We added the following to the discussion: Regarding future research, further knowledge is needed to explore feasibility and (cost-)effectiveness of I-MANAGE, as it has not yet been proven. Since the programme is very context specific and can be tailored according to the needs and resources of an organisation, it would be beneficial to investigate what works for which target group and under what circumstances, for example by means of a realist evaluation [64].	
86. In the discussion about the COPM you only refer to positive findings but reference 35 indicate that COPM might be challenging to use “by the book” (e.g. including scoring) for 100% of all clients. Please consider if you like to frame it as you do in the results “Clients MUST score both their performance and their satisfaction with performing these activities”. What if some clients can’t engage in the scoring procedure? Our response: We thank the reviewer for this comment. We agree that for some clients it may be difficult to score both their performance and satisfaction with the activities mentioned. However, this is a requirement of the COPM and that is why we framed it as ‘must score’. Additionally, we do believe that there are multiple ways to help a client to score these activities and occupational therapists are also trained in these communication/motivation skills. However, we slightly adapted the wording in the results: This instrument requires the clients to score both their performance and satisfaction when performing these activities [46]. In the discussion we do not refer to positive findings regarding the COPM scores, we refer to literature that says that integrating a standardized assessment and/or goal-setting tool could increase the effectiveness of reablement interventions and increases client involvement, etc. The choice for the COPM was a result of previous	P 17

literature research and the co-design process with stakeholders.	
87. In the following sentence it seems as you know that I-MANAGE will be successful. Before preliminary testing and rigorous evaluation can you actually claim the following? "First, supporting informal caregivers also contributes to the success of I-MANAGE." I do agree that it is an important component of an intervention program! In addition, success is a strong word and if it is to be used I suggest the authors specify what they mean with success. Our response: We thank the reviewer for this comment and agree that this may be formulated too strongly since results or success has not yet been proven. Therefore, we adapted the text: Firstly, supporting informal caregivers is assumed to contribute to the effectiveness of I-MANAGE.	P 20 - 21

VERSION 3 – REVIEW

REVIEWER	Marianne Eliassen UiT The Arctic University of Norway Faculty of Health Sciences
REVIEW RETURNED	15-May-2023

GENERAL COMMENTS	Dear authors, You have done a great job on revising the manuscript in accordance with the remarks from the previous review. Things are getting clearer. I have struggled a bit with the reading of this manuscript and think I finally have found a 'main source' for the confusion. I believe that you are trying to do two things at the same time, which causes a level of conflict. First, you are trying to present a description of a universal reablement model that can be adjusted to varied contexts and settings. Additionally, you are trying to document and describe the rather specific process of the project you conducted within the Dutch context. The very precise descriptions of specific practices do not correspond very well with the aim of an adjustable model. In a way, I feel like this paper is sort of a hybrid between a research article and a project report. It is not very clear to the reader what can be adjusted for, and what needs to be part of the model to be able to refer to it as the I-MANAGE-model. For example, you have a section where you write about the practice-oriented training, and state that the reablement-team should receive a training before receding the program. I can understand that this was done in the context that you report on, but how should this be followed up in a general context? If a care facility in Thailand (or wherever in the world) were to implement an adjusted I-MANAGE program, who would provide the training, what should the training include? I believe that you report on the specific process of your project, without providing the theoretical and generalizing aspects that is needed for others to make a use of your work. The example above is only one of many, where you report very specific, but then add a sentence about how this can be adjusted according to other settings. You state that the reablement program
---

	should be 8 weeks, with a maximum of two weeks extension. You also state that all team members should have access to the client's electronic care file. What if the context requires different length of the model, or if the country does not have a well-established electronic file system? Is it still possible to implement an I-MANADGE program. It is not very clear what can be adjusted for, and what needs to be a minimum standard. Although this comment is a bit wage, and probably freaks you out, as this review have been going on for several rounds now, I think you can handle this issue with small adjustments to the wordings through out the paper: What are requirements of the model, and what are recommendations (and can be adjusted for in various settings)? If you manage to include this aspect in the paper, I can recommend this article for submission. Good luck!
--	---

REVIEWER	Magnus Zingmark Municipality of Östersund, Health and Social Care Administration
REVIEW RETURNED	08-Jun-2023

GENERAL COMMENTS	Sorry for the delay. I posted this review a few days ago bot something must have went wrong. I think the authors have addressed all previous concerns I had or responded in a good way arguing for choices they have made. I therefore consider the manuscript ready for acceptance.
--

VERSION 3 – AUTHOR RESPONSE

Response to concerns raised by reviewer 1	
Introduction	
You have done a great job on revising the manuscript in accordance with the remarks from the previous review. Things are getting clearer. I have struggled a bit with the reading of this manuscript and think I finally have found a 'main source' for the confusion. I believe that you are trying to do two things at the same time, which causes a level of conflict. First, you are trying to present a description of a universal reablement model that can be adjusted to varied contexts and settings. Additionally, you are trying to document and describe the rather specific process of the project you conducted within the Dutch context. The very precise descriptions of specific practices do not correspond very well with the aim of an adjustable model. In a way, I feel like this paper is sort of a hybrid between a research article and a project report. It is not very clear to the reader what can be adjusted for, and what needs to be part of the model to be able to refer to it as the I-MANAGE-model:  For example, you have a section where you write about the practice-oriented training, and state that the reablement-team should receive a training before receding the program. I can understand that this was done in the context that you report on, but how should this be followed up in a general context? If a care facility in Thailand (or wherever in the world) were to implement an adjusted I-MANAGE program, who would provide the training, what should the training include? I believe that you report on the specific process of your project, 	P 13-18 P 19-21

without providing the theoretical and generalizing aspects that is needed for others to make a use of your work.

The example above is only one of many, where you report very specific, but then add a sentence about how this can be adjusted according to other settings.

- *You state that the reablement program should be 8 weeks, with a maximum of two weeks extension. You also state that all team members should have access to the client's electronic care file. What if the context requires different length of the model, or if the country does not have a well-established electronic file system? Is it still possible to implement an I-MANAGE program. It is not very clear what can be adjusted for, and what needs to be a minimum standard.*

Although this comment is a bit vague, and probably freaks you out, as this review have been going on for several rounds now, I think you can handle this issue with small adjustments to the wordings throughout the paper: What are requirements of the model, and what are recommendations (and can be adjusted for in various settings)? If you manage to include this aspect in the paper, I can recommend this article for submission.

Our response:

We thank the reviewer for this comment. First, we would like to point out that when this manuscript was written, the project/ I-MANAGE programme was not yet implemented in the Dutch setting and the results are therefore not a project report. We choose to present the results as obtained from our development process (i.e. input from literature, context analysis, working group sessions, and individual interviews). The result is a reablement model that fits within context of the Netherlands. Nevertheless with adjustments it can also be of added value for countries that are more or less comparable to the Netherlands. To clarify which elements are essential and which elements can be adjusted, we added the following sentence to the beginning of the result section:

The I-MANAGE programme, as described here, is the result of input from stakeholders and literature research during the last step of the development process (step 7) [41].

Additionally, we checked and adjusted the wording in the result section and used, for example, words like 'should' for essential components and words like 'could' or 'preferably' for components that allow adjustment.

We later on report in the discussion section on the parts that are essential and the parts that could be adjusted to the needs and resources of a specific context (p 19-21).

However, we understand that some more explanation on what is essential and what could be adjusted can be useful for the reader. We therefore added the some

sentences for clarification throughout the discussion section: The programme contains several key elements that are considered essential and should be present when implementing the programme in any care setting. Firstly, in line with the conceptual definition of reablement [18], interdisciplinary collaboration is important in I-MANAGE. However, how this element is implemented in practice depends on the contextual circumstances of a country or region. In this study it was recommended by Dutch stakeholder to appoint a care coordinator, schedule (bi-)weekly meetings, and implement a shared ECF. Additionally, the integrated practice-oriented and on the job training, where care professionals can learn from other disciplines, help to invest in the self-efficacy of care professionals. This is essential, because successfully changing behaviours remains a challenge. The training entails several key topics as mentioned before, however, depending on the local context the extend of the training may vary, for example, due to previously received education or training. Previous research has indicated that using standardised assessment or goal-setting tools could increase the effectiveness of reablement interventions, and is therefore considered an essential element of the programme [30]. Also, the duration and intensity of the programme may vary according to the needs of the local context and chosen target population. Moe and Brinchmann [25] confirmed the necessity of tailoring reablement services to local conditions by arguing that establishing reablement in an existing organisational structure is a complex process.	
Response to concerns raised by reviewer 2	
1. I think the authors have addressed all previous concerns I had or responded in a good way arguing for choices they have made. I therefore consider the manuscript ready for acceptance. Our response: We thank the reviewer for this comment.	N/A

VERSION 4 – REVIEW

REVIEWER	Marianne Eliassen UiT The Arctic University of Norway Faculty of Health Sciences
REVIEW RETURNED	24-Jun-2023
GENERAL COMMENTS	Thank you for being patient with me and all my input on this manuscript. I'm glad you persevered, as the article now appears as well-written, interesting and relevant. The changes made in the last round of review clarify how the I-MANAGE model can be used in

	different settings, despite the fact that it was developed in, and adapted to, the Dutch context. Brilliant.
--	--